# Cytochrome c oxidase dependent respiration is essential for T cell activation, proliferation and memory formation

Tatiana N. Tarasenko[1], Emily Warren [1,5], Bharati Singh[1], Amanda Fuchs [1], Jose Marin [1], Marten Szibor [2,3], Christopher King [4] & Peter J. McGuire [1] ✉

T cell activation requires extensive metabolic reprogramming, but the specific requirement for mitochondrial respiration (MR) remains unresolved. While most studies have focused on aerobic glycolysis as the primary driver of proliferation and effector function, the role of MR has not been completely defined. To isolate MR from proton pumping by cytochrome c oxidase (COX), we expressed the non-proton-pumping alternative oxidase (AOX) in activated COX-deficient T cells. AOX restored electron flow, membrane potential, and mitochondrial ATP production, ultimately rescuing proliferation, effector and memory differentiation, and antiviral immunity. These improvements required upstream electron input, particularly from Complex I, with Complex II and DHODH contributing more modestly. Despite restored MR, glycolysis remained elevated, likely due to altered redox signaling. These findings demonstrate that MR, normally mediated by COX, is necessary and can be sufficient to support T cell activation and function, independent of proton translocation, provided upstream electron input is maintained.

Unlike most somatic cells, immune cells are not terminally differentiated and retain the capacity to undergo metabolic reprogramming upon activation, a process that is essential for driving proliferation and differentiation into effector and memory states. T cell activation triggers a marked increase in glycolytic flux to meet the elevated energetic and biosynthetic demands[1] of proliferation by supplying ATP and key metabolic intermediates for nucleotides, amino acids, and lipids[2,3]. This metabolic shift is a prerequisite for the transition from activation to biomass accumulation and cell cycle progression, effectively supporting clonal expansion.

Mitochondrial oxidative phosphorylation (OXPHOS) is also induced following T cell activation and persists during proliferation. OXPHOS encompasses the electron transport chain (ETC), oxygen consumption via respiration, and ATP synthesis driven by the resulting proton gradient. Although both glycolysis and OXPHOS are

upregulated, the latter may produce outputs that are not redundant with glycolytic metabolism. Indeed, individual ETC complexes have distinct roles in T cell function. For instance, complex III promotes T cell activation through the regulated generation of reactive oxygen species[4] and is required for suppressive function in regulatory T cells[5]. Previous work from our group demonstrated that intact complex IV, cytochrome c oxidase (COX), prevents apoptosis following activation and is required for T cell proliferation, differentiation, and immune function[6].

COX, the terminal complex of the ETC, catalyzes the reduction of molecular oxygen to water. This reaction is driven by electrons donated from cytochrome c and proceeds through a series of redox centers, including two heme groups and two copper centers[7,8]. The terminal step of this reaction, in which oxygen is reduced to water, constitutes MR. Coupled to this redox reaction is the active pumping

[1]Metabolism, Infection, and Immunity Section, National Human Genome Research Institute, National Institutes of Health, Bethesda, MD, USA. [2]Faculty of Medicine and Health Technology, Tampere University, Tampere, Finland. [3]Department of Cardiothoracic Surgery, Center for Sepsis Control and Care (CSCC), Jena University Hospital, Jena, Germany. [4]Division of Veterinary Recourses, National Institutes of Health, Bethesda, MD, USA. [5]Deceased: Emily Warren. ✉e-mail: peter.mcguire@nih.gov

of protons across the inner mitochondrial membrane, thereby contributing to the proton motive force. As such, COX supports several higher-order processes in T cells, including mitochondrial DNA maintenance, transcription, and resistance to cytochrome c−mediated apoptosis[6,9]. However, the specific contributions of MR versus proton pumping to these outcomes remain unresolved.

Although glycolysis has been the primary focus during metabolic reprogramming in T cell activation, evidence from models such as rho[0] cells, which lack mitochondrial DNA and therefore functional MR, show impaired proliferation[10] and differentiation capacity[11]. These deficits point to mitochondrial outputs, including those linked to MR, as potential regulators of critical T cell functions. While our previous work demonstrated that COX deficiency disrupts proliferation, survival, and immune function[6], the specific role of MR in these outcomes remains unresolved. We hypothesized that the respiratory activity of COX, independent of its proton-pumping function, is both necessary and sufficient to drive downstream expansion, differentiation, and effector function. To directly test this, we expressed the alternative oxidase (AOX) from *Ciona intestinalis*[12,13] in COX-deficient T cells. AOX transfers electrons from ubiquinol to oxygen without translocating protons, allowing us to study MR without COX-mediated proton pumping. This system enabled us to isolate the contribution of MR to T cell metabolic reprogramming and to assess its sufficiency in supporting key functional outcomes. More broadly, our findings highlight that MR, though often overshadowed by glycolysis, is a central determinant of T cell fate and function.

## Results

### AOX restores cell cycle, apoptosis, and differentiation transcriptional pathways

COX10, a protoheme:heme-O-farnesyl transferase, is indispensable for the biosynthesis of heme a, the prosthetic group of COX. Deficiency of COX10 results in marked impairment of COX and OXPHOS[14]; the molecular pathology of our previously published model in T cells (*TCox10−/−*)[6]. To restore MR specifically, we introduced a ubiquinol oxidase (AOX) from *Ciona intestinalis* into *TCox10−/−* T cells. We engineered this mouse model by breeding *TCox10−/−* mice with counterparts constitutively expressing the *Aox* gene (*Gt(ROSA)26Sortm1.1(CAG-AOX)Htj*)[12] to generate *TCox10−/−/Aox* mice (Fig. 1A), with PCR and qPCR analysis confirming *Aox* expression and *Cox10* deletion in the progeny (Fig. 1B, C). As AOX is an electron acceptor upstream of complex III (Fig. 1D), its activity depends on electron input into CoQ from Complex I, Complex II, and dihydroorotate dehydrogenase (DHODH). AOX-expressing cells can resist the effects of sodium azide, a toxin for COX, and continue respiration unaffected[15–17]. T cells derived from WT, *Aox*, *TCox10−/−*, and *TCox10−/−/Aox* mice were activated for 24 h with anti-CD3/CD28 and oxygen consumption rates (OCR) was measured after sodium azide port injection. AOX-expressing cells maintained elevated OCR despite COX inhibition[15], in contrast to the expected decline in WT and *TCox10−/−* T cells (Fig. 1E).

To understand how AOX may impact cellular dynamics, we performed RNAseq on WT, *Aox*, *TCox10−/−*, and *TCox10−/−/Aox* T cells stimulated as above. Each genotype showed significant changes in gene expression relative to WT, or in *TCox10−/−/Aox* against *TCox10−/−* (Supplementary Fig. 1A and Supplementary Data 1). As these comparisons suggested considerable changes in gene expression across the four genotypes, we performed a weighted gene coexpression network analysis (WGCNA) to simultaneously compare them. Hierarchical clustering grouped the genes into 18 modules (Supplementary Fig. 1B). Examining the correlation of each module's eigengene intercepts allowed us to identify which modules had gene expression perturbed in *TCox10−/−* and normalized by the introduction of AOX (Supplementary Fig. 1C). We selected four modules (turquoise, yellow, greenyellow, and midnightblue) that followed this pattern and used a heatmap of the normalized gene expression from each

category to confirm the differences in expression across groups (Supplementary Fig. 1D). Overrepresentation analysis (ORA) of each of the modules indicated the main functions of the modules in these gene sets (Supplementary Data 2), which included apoptotic signaling, cell cycle phase transition, and T cell proliferation, differentiation, and function (Fig. 1F). These data establish that AOX reprograms gene expression to support T cell fitness and function.

### AOX improves mitochondrial structure and redox homeostasis in activated T cells

Building on the broad transcriptomic correction observed with RNA-seq, we next focused on whether AOX specifically influences mitochondrial pathways. To address this, we performed focused transcriptomic analysis of *TCox10−/−/Aox* cells. Genes involved in OXPHOS, particularly those encoding Complex I subunits, were re-enriched (Supplementary Fig. 2). This observation was particularly important because, in this context, Complex I is the sole contributor to proton pumping in our model (Fig. 1D). Additional transcriptional improvements were observed in pathways related to mtDNA metabolism, mitochondrial translation, and nucleotide and carbohydrate metabolism (Supplementary Fig. 2 and Supplementary Data 3), indicating broad restoration of mitochondrial respiratory and biosynthetic programs.

Mitochondrial morphology is intimately tied to bioenergetic function in cells[18]. We next asked whether the transcriptional shifts observed were accompanied by structural recovery of mitochondria. Electron microscopy of T cells activated for 24 h revealed that *TCox10−/−* cells exhibited elongated mitochondria of similar width with increased surface area and reduced number per field (Fig. 2A–E). This mitochondrial elongation likely reflects a compensatory response to respiratory chain dysfunction, as observed in other COX-deficient cell types[19,20]. These abnormalities were reversed in *TCox10−/−/Aox* cells, whose morphology resembled that of WT. Despite these corrections, mtDNA content remained elevated in both *TCox10−/−* and *TCox10−/−/Aox* cells (Fig. 2F), suggesting a persistent compensatory response to COX loss.

Given these improvements in mitochondrial architecture, we next evaluated whether AOX also restored aspects of mitochondrial function and redox homeostasis. *TCox10−/−* T cells displayed a trend in elevated mitochondrial membrane potential ($\Delta\Psi m$) with TMRE (Fig. 2G), seen more in CD4[+] T cells, consistent with impaired electron transport[21,22]. Given the reduced mitochondrial number in *TCox10−/−* cells, the TMRE signal likely underestimates the extent of mitochondrial hyperpolarization on a per-organelle basis. AOX expression improved membrane potential, approximating WT levels. Redox imbalance was also present in *TCox10−/−* cells, marked by elevated total ROS, reduced superoxide, and increased hydrogen peroxide by flow cytometry (Fig. 2H–J), suggesting increased superoxide dismutase activity. In *TCox10−/−/Aox* cells, total ROS and hydrogen peroxide returned to normal, while superoxide remained reduced. This persistent reduction in superoxide may reflect the metabolic bypass of complex III by AOX (Fig. 1D). The changes in redox balance were further supported by restoration of the $NAD^+$/NADH ratio in *TCox10−/−/Aox* cells (Fig. 2K).

### AOX-MR uncovers complex I reserve capacity and metabolic flexibility in activated T cells

Although MR increases during T cell activation, the requirement for specific electron inputs into the ETC and their contribution to cellular function remains to be defined. Given that AOX restored mitochondrial structure and improved redox balance in COX-deficient T cells, we next confirmed the reestablishment of functional MR in 24 h activated T cells. Extracellular flux analysis revealed markedly increased OCR in activated CD4[+] and CD8[+] *TCox10−/−/Aox* cells, exceeding those

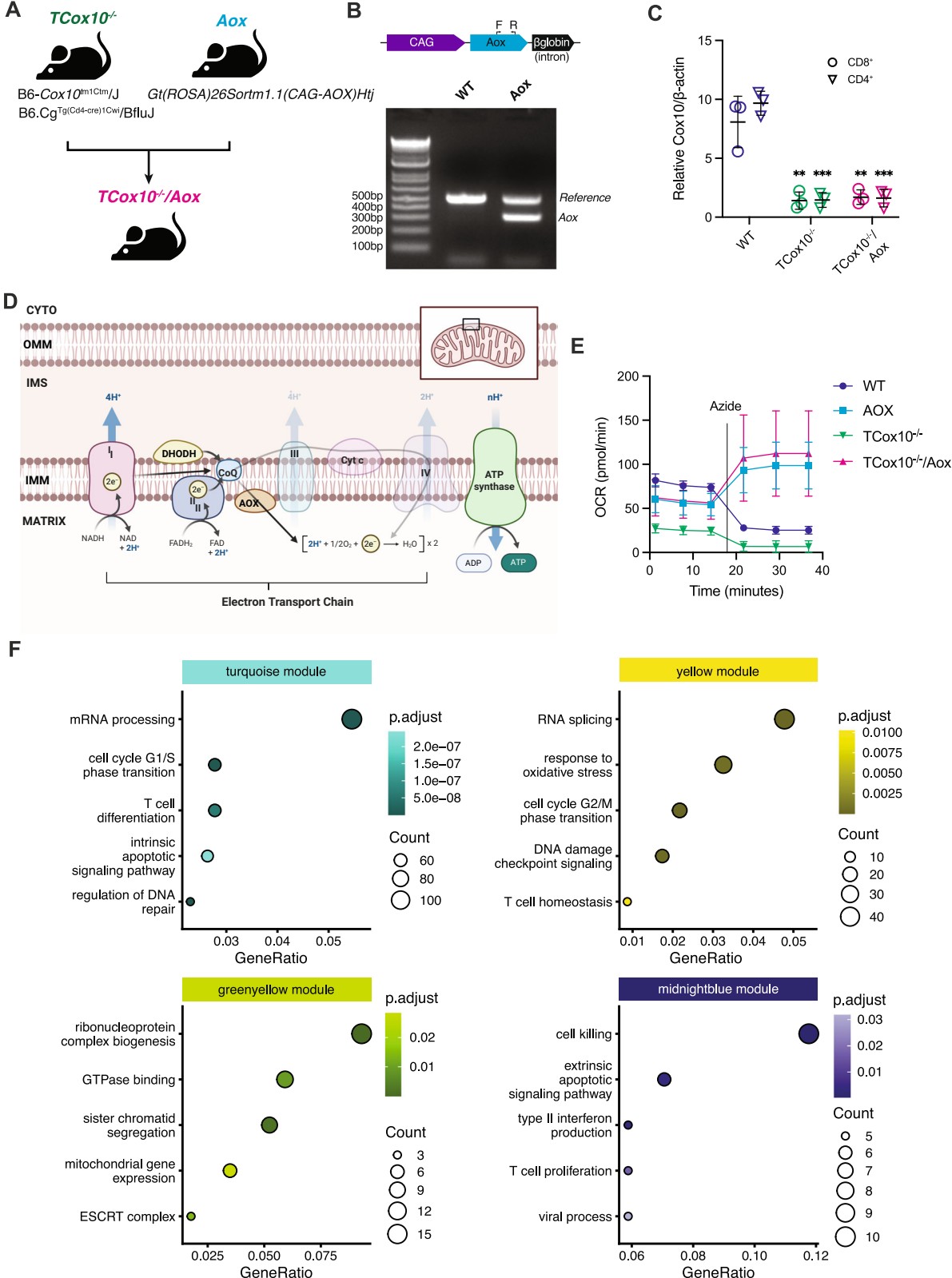

of WT (Fig. 3A), consistent with a restoration of MR. To determine whether COX activity was enhanced by AOX, we used N′-tetramethyl-para-phenylene-diamine (TMPD), a direct substrate for Complex IV. COX-MR remained minimal in *TCox10⁻/⁻/Aox* cells as expected (Fig. 3B), confirming that the observed OCR increase was COX-independent and mediated by AOX. AOX expression also improved mitochondrial ATP levels as measured by ATP-Red and flow cytometry (Fig. 3C).

Because ATP synthesis ultimately depends on proton gradient maintenance, we next asked how AOX-MR supports ΔΨm in the absence of COX. Since AOX does not pump protons, its ability to support ΔΨm depends on Complex I. To test this, we measured ΔΨm by TMRE staining in activated CD4⁺ and CD8⁺ T cells treated with inhibitors targeting distinct electron donors to the CoQ pool (Fig. 3D). Complex I inhibition with piericidin led to a sharp loss of TMRE across

**Fig. 1 | AOX restores cell cycle, apoptosis, and differentiation transcriptional pathways. A** Schematic of mouse breeding scheme to generate *TCox10⁻/⁻/Aox* mice. In brief, mice with CD4-Cre driven knockout of *Cox10* (*TCox10⁻/⁻* mice) were crossed with mice with AOX expression to generate *TCox10⁻/⁻/Aox* mice. Created in BioRender. Mcguire, P. (2025) https://BioRender.com/ljr6y8o. **B** Schematic of *Aox* transgene (top) and PCR confirmation of successful expression of *Aox*. **C** qPCR quantification of genomic DNA of *Cox10* gene in both *TCox10⁻/⁻* and *TCox10⁻/⁻/Aox* mice (*N* = 3/condition). CD8⁺ and CD4⁺ spleen cells were sorted, gDNA was extracted and relative level of *Cox10* was determined by quantitative PCR relative to β-actin level. **D** Schematic of mitochondrial respiratory chain with exogenous expression of alternative oxidase (AOX). Created in BioRender. McGuire, P. (2025) https://BioRender.com/ss9r7kr. In the absence of complex IV activity (*TCox10⁻/⁻* mice). AOX accepts electrons passed from complexes I and II to generate water and maintain proton motive force for ATP production. CYTO cytoplasm, OMM outer

mitochondrial membrane, IMS inner membrane space, IMM inner mitochondrial membrane, MATRIX mitochondrial matrix, DHODH dihydroorotate dehydrogenase, Coq coenzyme Q, AOX alternative oxidase, Cyt c cytochrome c. **E** Extracellular flux analysis of oxygen consumption rate (OCR) in T cells following treatment with 0.25 mM sodium azide. Pan T cells were isolated by magnetic beads and stimulated for 24 h with plate bound anti-CD3 and anti-CD28. Cells were attached to Seahorse plates (2 × 10⁵ cells/well) and 0.25 mM sodium azide was injected. Representative graph shown. (*N* = 7–8/condition). **F** Overrepresentation analysis of four WGCNA modules. Selected enriched pathways are shown for each module (*N* = 3–5/condition). Point color reflects B-H adjusted *p*-value, size reflects enriched genes in the set. Data are representative of two to three independent experiments and indicate mean and standard deviation. * $p < 0.05$, ** $p < 0.01$, *** $p < 0.001$ by one-way ANOVA and post-hoc Dunnett test against WT.

all genotypes, including AOX-expressing cells, confirming that ΔΨm in these cells depends on Complex I-driven proton pumping. Although Complex II and DHODH do not pump protons, they supply electrons to CoQ (Fig. 1D) and could thereby help sustain a reduced CoQ pool. Inhibition of Complex II with TTFA reduced TMRE signal in all groups, although the effect was less pronounced than with piericidin. This suggests that Complex II can play a supportive role in maintaining ΔΨm by sustaining electron flow into the CoQ pool. In contrast, DHODH inhibition with brequinar caused an even more modest reduction in TMRE, indicating that pyrimidine-linked electron flow makes a smaller contribution under these conditions.

To determine whether the same electron sources that support ΔΨm also sustain ATP synthesis, we measured mitochondrial ATP levels following treatment with the same inhibitors in activated CD4⁺ and CD8⁺ T cells (Fig. 3E). Inhibition of Complex I with piericidin led to a pronounced reduction in ATP across all genotypes, confirming the essential role of Complex I-driven proton pumping in supporting ATP production. TTFA treatment caused a moderate decline in ATP levels in AOX-expressing cells, while brequinar had even less of an effect. These findings suggest that Complex II and DHODH play a minor supportive role in sustaining mitochondrial ATP synthesis, likely via maintaining the reduced state of the CoQ pool and ETC flow.

With improved MR, ΔΨm, and ATP synthesis with AOX-MR, we next asked whether it also corrected metabolic rewiring; in particular, the glutamine dependence previously observed in COX-deficient T cells[6]. Glutamine is an anaplerotic amino acid in the TCA cycle, generating reducing equivalents that can drive OXPHOS[23,24]. To evaluate whether AOX reverses this metabolic shift, we performed ¹³C-glutamine tracing in activated T cells (Fig. 3F). *TCox10⁻/⁻/Aox* cells showed reduced incorporation of ¹³C-glutamine into downstream TCA intermediates fumarate, malate, and aspartate (all M + 4), restoring levels similar to WT (Fig. 3G). Citrate labeling (M + 2 and M + 4) also improved (Fig. 3H), consistent with restored TCA cycle dynamics. These findings suggest that MR is required not only for ATP production but also modulates metabolic flexibility during T cell activation, enabling adaptive substrate use.

### Glycolysis remains elevated with AOX-MR in COX-deficient T cells

The TCA cycle and glycolysis are integral to T cell metabolic reprogramming following activation[3]. Although AOX restored MR, membrane potential, and ATP synthesis in COX-deficient T cells, it remained unclear whether this recovery was sufficient to normalize glycolysis, which is typically upregulated as a compensatory response to impaired OXPHOS[25]. Extracellular flux analysis revealed that extracellular acidification rates (ECAR) in activated *TCox10⁻/⁻/Aox* T cells remained high and comparable to *TCox10⁻/⁻* cells, indicating sustained glycolytic activity despite restored respiration (Fig. 4A). To determine whether the elevated ECAR reflected lactate production rather than alternative acidifying processes, we treated cells with an LDH inhibitor. The

resulting reduction in ECAR suggested that acidification was primarily due to lactate in *TCox10⁻/⁻/Aox* cells (Fig. 4B). Direct quantification of lactate in the media further supported this finding (Fig. 4C).

To test whether elevated glycolysis in *TCox10⁻/⁻/Aox* T cells was driven by HIF1α stabilization, we measured HIF1α by flow cytometry (Fig. 4D). While HIF1α was elevated in *TCox10⁻/⁻* T cells, levels were normalized in *TCox10⁻/⁻/Aox* cells, indicating that persistent glycolysis in the rescued cells occurs through a HIF1α-independent mechanism. This observation was unexpected, as AOX restored redox balance and prevented mtROS production. AOX also bypasses Complex III, which is a site of redox signaling, and mitochondrial superoxide remained low in *TCox10⁻/⁻/Aox* cells (Fig. 2I). These findings raise the possibility that altered redox signaling contributes to the persistent glycolysis, although other mechanisms may also be involved.

To determine whether AOX-MR enables glucose entry into oxidative pathways, we performed ¹³C-glucose tracing (Fig. 4E). *TCox10⁻/⁻/Aox* cells showed increased incorporation of ¹³C-glucose into both lactate and pyruvate compared to *TCox10⁻/⁻* cells (Fig. 4F). The reduced labelling seen in *TCox10⁻/⁻* is likely due to lactate export by the monocarboxylate transporter[26], as reflected in our extracellular flux analysis and lactate measurements. ¹³C-labeled glucose was also detected in multiple TCA cycle intermediates (Fig. 4G), along with citrate labeling patterns (M + 2 and M + 4) consistent with active cycling of the TCA (Fig. 4H), indicating that glucose-derived carbon enters, feeds, and recirculates through the cycle in AOX-expressing cells.

### AOX-MR abrogates apoptosis in activated T cells

MR is closely linked to apoptotic regulation, and our previous work showed that COX deficiency promotes apoptosis during the proliferative phase following T cell activation[6]. Given that AOX-MR restored key metabolic features of activated T cells, we next asked whether this was sufficient to rescue cellular survival. Building on our previous findings, alterations in apoptosis were disentangled by mapping *TCox10⁻/⁻/Aox* versus *TCox10⁻/⁻* log₂ fold changes from our RNAseq onto the KEGG apoptosis pathway (mmu04210) (Supplementary Fig. 3A). While multiple pro-apoptotic and pro-survival genes were upregulated in *TCox10⁻/⁻* T Cells, interestingly, elevated expression of extrinsic activators *Fas*, *FasL*, *Perf1*, and *Tradd* was reversed by AOX. Apoptosis was measured in T cells by live/dead dye and Annexin V staining at 72 h post-activation (Fig. 5A), revealing an improvement in *TCox10⁻/⁻/Aox* CD4⁺ and CD8⁺ T viability (Fig. 5B). To investigate potential mechanisms underlying the reduction in apoptosis, we assessed caspase 3 activation, a common pathway for both intrinsic and extrinsic apoptosis. Caspase 3 activation was abnormally elevated in *TCox10⁻/⁻* T cells and decreased in *TCox10⁻/⁻/Aox* T cells (Fig. 5C). We also observed similar trends in specific apoptotic pathways, showing amelioration of both caspase 8 (i.e., extrinsic pathway, Fig. 5C) and caspase 9 (i.e., intrinsic pathway, Fig. 5C) activation in *TCox10⁻/⁻/Aox*

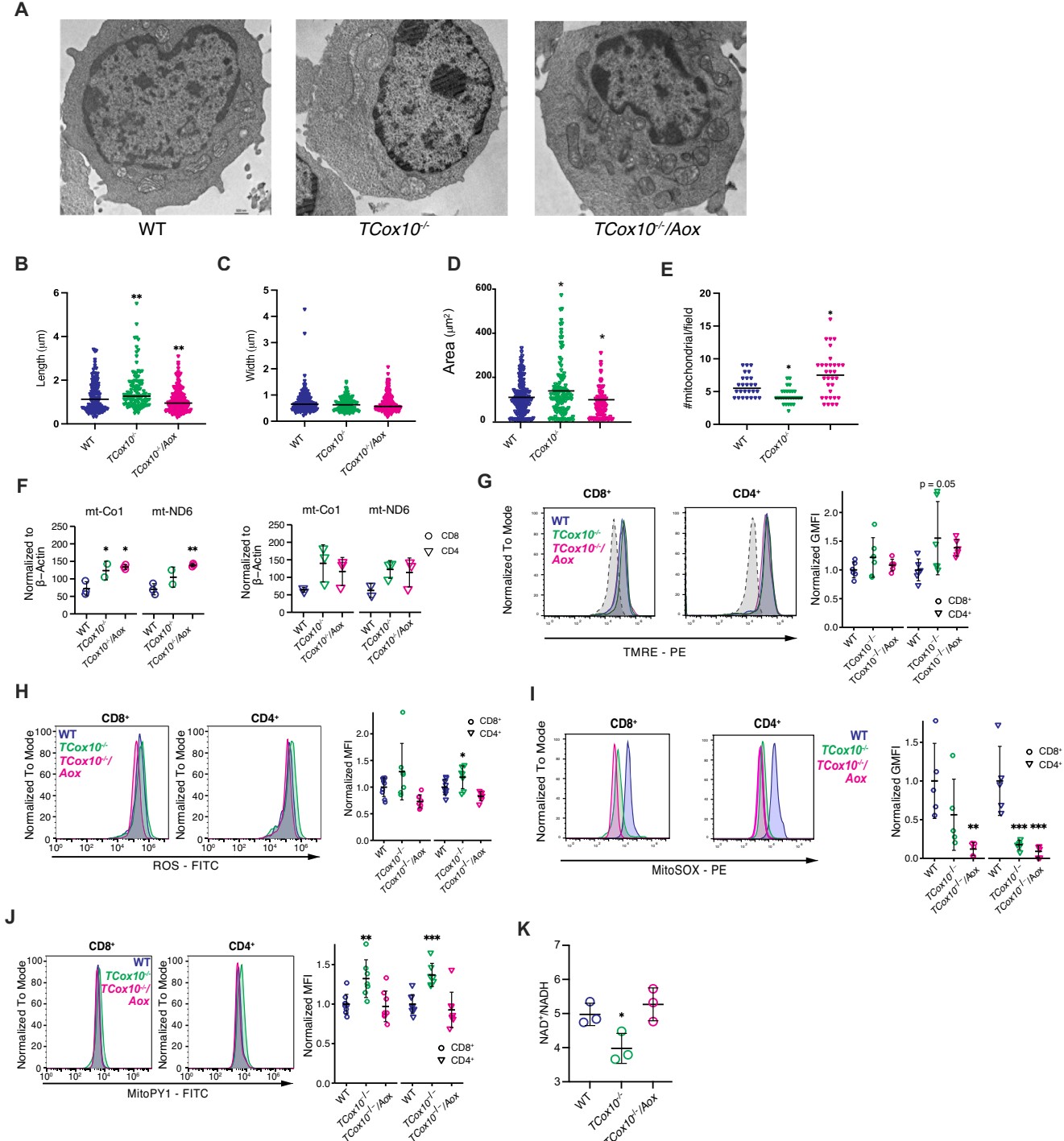

**Fig. 2 | AOX improves mitochondrial structure and redox homeostasis in activated T cells.** Splenic T cells were stimulated for 24 h as above.
**A** Representative Transmission electron micrograph of WT, *TCox10⁻/⁻* and *TCox10⁻/⁻/Aox* T cells. Quantification of **B** mitochondrial length, **C** width, **D** area, and **E** number of mitochondria per field ($N = 3$/condition). **F** qPCR quantification of mtDNA copy number normalized to β-actin ($N = 3$/condition). Genomic DNA from sorted CD4⁺ and CD8⁺ splenocytes was isolated and qPCR was performed. CD8⁺cells left, CD4⁺ right. **G** Normalized mean fluorescence intensity (MFI) of TMRE in activated T cells ($N = 6$–7/condition). **H** Total ROS fluorescence in activated CD8⁺ and CD4⁺ T cells ($N = 7$–9/condition). Left, representative density plot; right, quantification of normalized GMFI. **I** MitoSOX fluorescence in activated CD8⁺ and CD4⁺ T cells ($N = 5$–9/condition). Left, representative density plot; right, quantification of normalized MFI. **J** MitoPY1 fluorescence in CD8⁺ and CD4⁺ T cells ($N = 7$–9/condition). Left, representative density plot; right, quantification of normalized MFI. **K** NAD⁺/NADH ratio in T cells by HPLC ($N = 3$/condition). Data are representative of 2–3 experiments and indicate mean and standard deviation. Statistical significance is indicated by asterisks; ** $p < 0.01$, * $p < 0.05$.

T cells. Since we observed increased expression of Caspase 8 and given our RNAseq results (Supplementary Fig. 3A and Supplementary Data 1), we further examined activation of the extrinsic pathway by quantifying Fas and FasL (Fig. 5D, E). Expression of both proteins were

increased in *TCox10⁻/⁻* T cells and reduced with AOX-MR. To probe the activity of this pathway in vitro, we used anti-FasL antibodies on WT and *TCox10⁻/⁻* CD8⁺ and CD4⁺ T cells. This intervention significantly increased the viability of *TCox10⁻/⁻* T cells (Supplementary Fig. 3B),

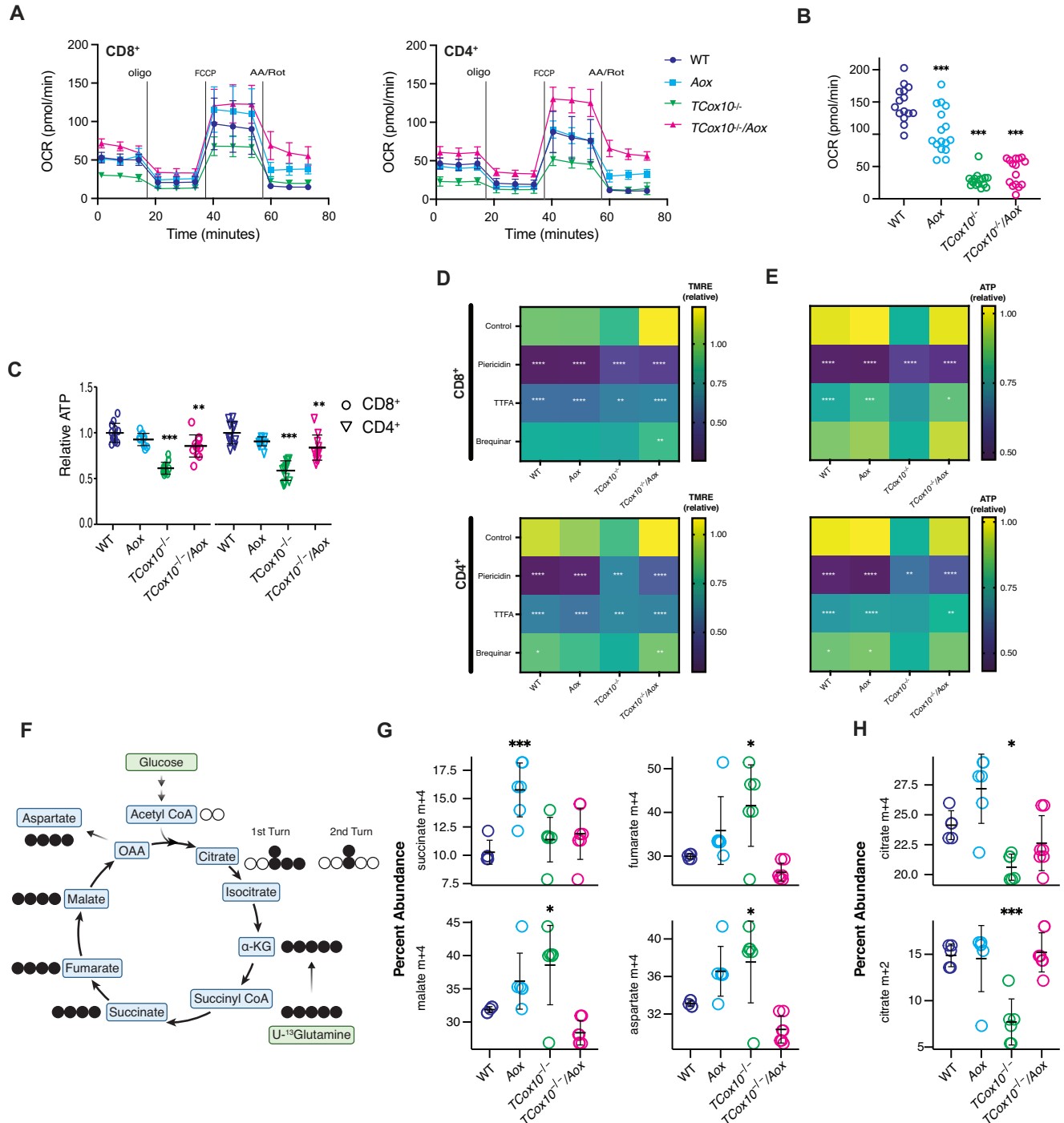

**Fig. 3 | AOX-MR uncovers complex I reserve capacity and metabolic flexibility in activated T cells.** Splenic T cells were isolated and stimulated for 24 h as above. **A** Oxygen consumption rate of CD8+ and CD4+ cells across genotypes ($N = 11–12/$ condition). Representative OCR plots. **B** Complex IV respiration in permeabilized T cells measured by N′-tetramethyl-para-phenylene-diamine (TMPD) oxidation ($N = 15–16/$condition). **C** Relative level of mitochondrial ATP measured by ATP-red staining by flow cytometry ($N = 6/$condition). **D** Heat map for TMRE for activated T cells treated with inhibitors of ETC inputs: piericidin (0.2 µM, complex I), thenoyltrifluoroacetone (50 µM, TTFA, complex II), brequinar (0.2 µM, DHODH, dihydroorotate dehydrogenase). **E** Heat map of relative ATP levels by ATP-Red and flow cytometry. Same inhibitors as (**D**). **F** Schematic of stable isotope labeling of TCA metabolites from U-13glutamine. Black circles indicate labeled carbons, white circles indicate unlabeled carbons. Activated T cells were incubated for 24 h with 2 mM U-13glutamine. Created in BioRender. Mcguire, P. (2025) https://BioRender.com/3vxnzzg. **G** Quantification of succinate, fumarate, malate, and aspartate (m + 4) abundance across genotypes ($N = 5–7/$condition). **H** Quantification of m + 4 citrate (first TCA turn) and m + 2 citrate (second TCA turn) abundance across genotypes ($N = 5–7/$condition). Data are representative of 3 experiments and indicate mean and standard deviation. * $p < 0.05$, ** $p < 0.01$, *** $p < 0.001$ by one-way ANOVA and post-hoc Dunnett test against WT.

indicating a major role for this pathway in COX-mediated apoptosis. However, blocking FasL did not improve proliferation as reflected by the retention of Cell Trace Violet (CTV, Supplementary Fig. 3C), underscoring the importance of maintaining MR for this function.

## AOX-MR sustains T cell function in vitro

T cell activation requires not only survival, but also robust proliferation and effector differentiation, processes that are tightly coupled to metabolic reprogramming[27]. Having shown that AOX-MR suppresses

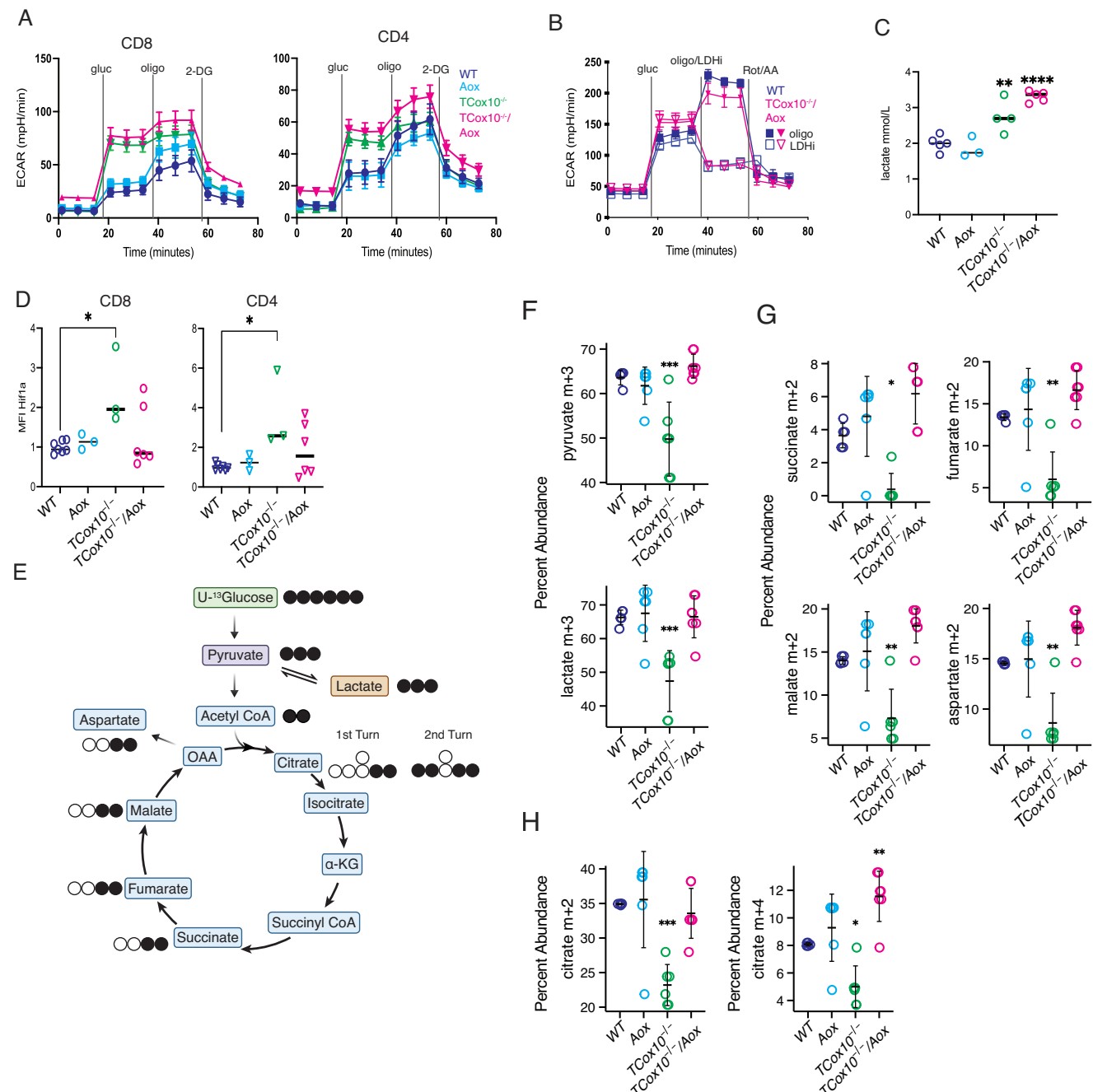

**Fig. 4 | Glycolysis remains elevated with AOX-MR in COX-deficient T cells.** Splenic T cells were isolated and stimulated for 24 h as above. **A** Measurement of extracellular acidification rate (ECAR) in CD8⁺ and CD4⁺ T cells (2 × 10⁵ cells/well, $N = 11–12$/condition). Representative ECAR plots. **B** ECAR in Pan T cells after injection of oligomycin (filled) or LDHA inhibitor (50 mkM, empty). **C** Lactate in the medium of activated T cells (2 × 10⁶ cells/ml) stimulated for 24 h ($N = 3–5$/condition). Lactate was measured by YSI analyzer in triplicate and averaged. Basal concentration was deducted from total production. **D** Hif1a staining by flow cytometry ($N = 3–7$/condition). **E** Schematic of stable isotope labeling with U-¹³glucose. Black circles indicate labeled carbons, white circles indicate unlabeled carbons. Created in BioRender. Mcguire, P. (2025) https://BioRender.com/zkz0xgp. **F** Quantification of pyruvate and lactate (m + 3) abundance deriving from glucose ($N = 5–7$/condition). **G** Quantification of succinate, fumarate, malate, and aspartate (m + 2) abundance deriving from glucose ($N = 5–7$/condition). **H** Quantification of citrate (m + 2) and citrate (m + 4) abundance deriving from glucose ($N = 5 = 7$/condition). Data are representative of two to three independent experiments and indicate mean and standard deviation. * $p < 0.05$, ** $p < 0.01$, *** $p < 0.001$ by one-way ANOVA and post-hoc Dunnett test against WT.

apoptosis in COX-deficient T cells, we next asked whether it was sufficient to support functional activation. We began by examining the impact of AOX-MR on proliferation, 72 hours following activation. *TCox10⁻/⁻/Aox* T cell proliferation matched WT, even when challenged with sodium azide (Fig. 6A). The division index further indicated that the AOX-MR could support proliferation (Fig. 6B). To determine which electron inputs were necessary to sustain AOX-MR supported

proliferation, we treated *TCox10⁻/⁻/Aox* cells with inhibitors targeting Complex I (piericidin), Complex II (TTFA), or DHODH (brequinar). Each intervention significantly impaired proliferation (Fig. 6C), indicating that sustained AOX-MR-supported proliferation requires Complex I, with Complex II and DHODH playing supportive roles in biosynthesis and maintaining CoQ redox balance and electron flow. We next examined whether restored proliferation in *TCox10⁻/⁻/Aox* cells was

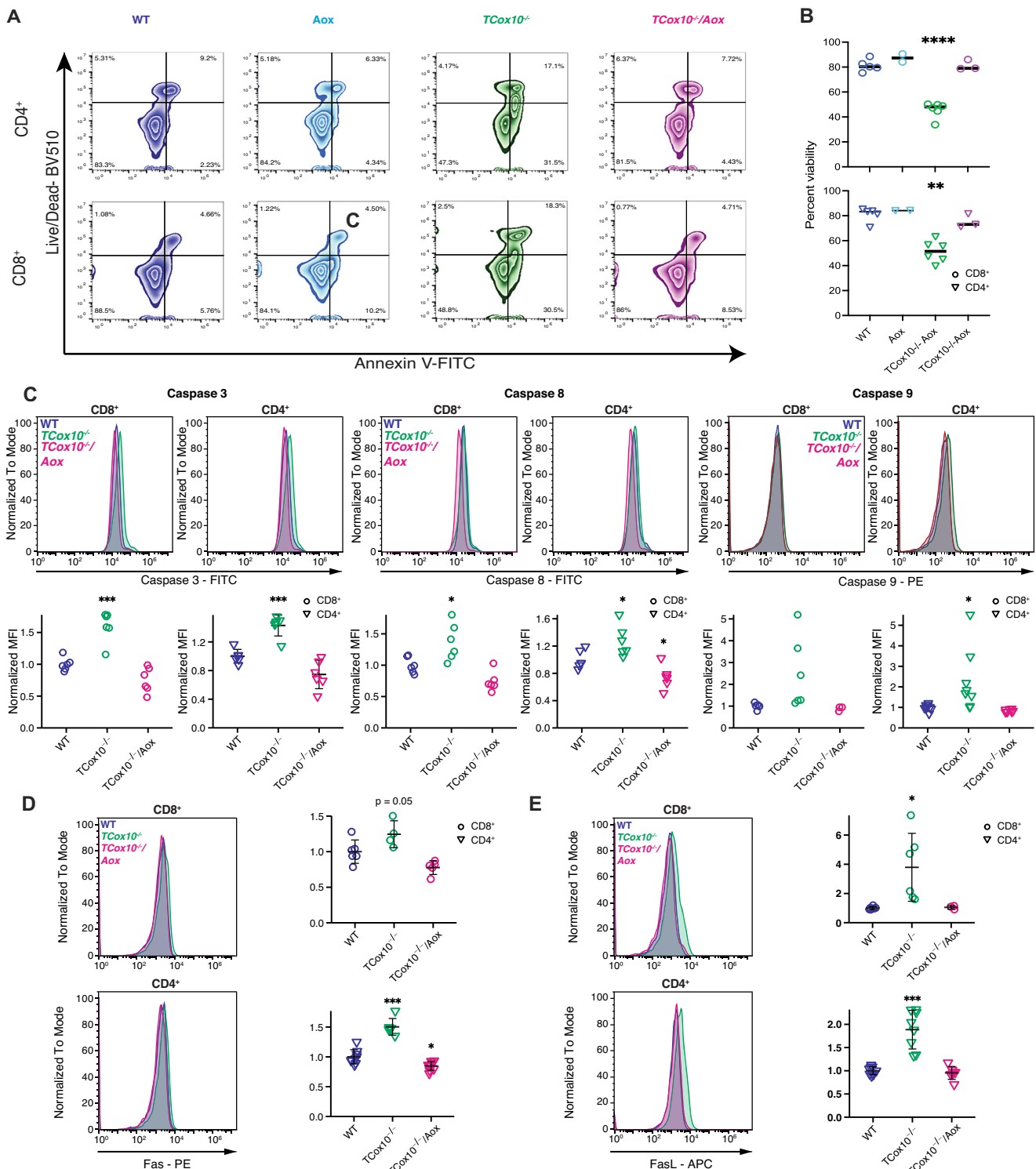

**Fig. 5 | AOX-MR abrogates apoptosis in activated T cells.** Splenic T cells were stimulated for 72 h with plate bound anti-CD3 and anti-CD28 antibodies as above. **A** Representative contour plot of viability analysis in CD8+ and CD4+ T cells using Live/Dead and Annexin V stains. **B** Quantification of viability analysis ($N = 3–6$/condition). **C** Caspase 3 (left), caspase 8 (center), and caspase 9 (right) activity in CD8+ and CD4+ T cells ($N = 6–9$/condition). Top, representative density plot; bottom, quantification of normalized mean fluorescence intensity (MFI). **D** Fas expression in CD8+ and CD4+ T cells ($N = 10–15$/condition). Left, representative density plot; right, quantification of normalized MFI. **E** FasL expression in CD8+ and CD4+ T cells ($N = 10–15$/condition). Left, representative density plot; right, quantification of normalized MFI. Cells were isolated by magnetic beads and stimulated for 72 h (for **A**) and 24 h for (**B**–**E**) with anti-CD3 and anti-CD28 in complete media. Data are representative of two to three independent experiments and indicate mean and standard deviation. * $p < 0.05$, ** $p < 0.01$, *** $p < 0.001$ by one-way ANOVA and post-hoc Dunnett test against WT.

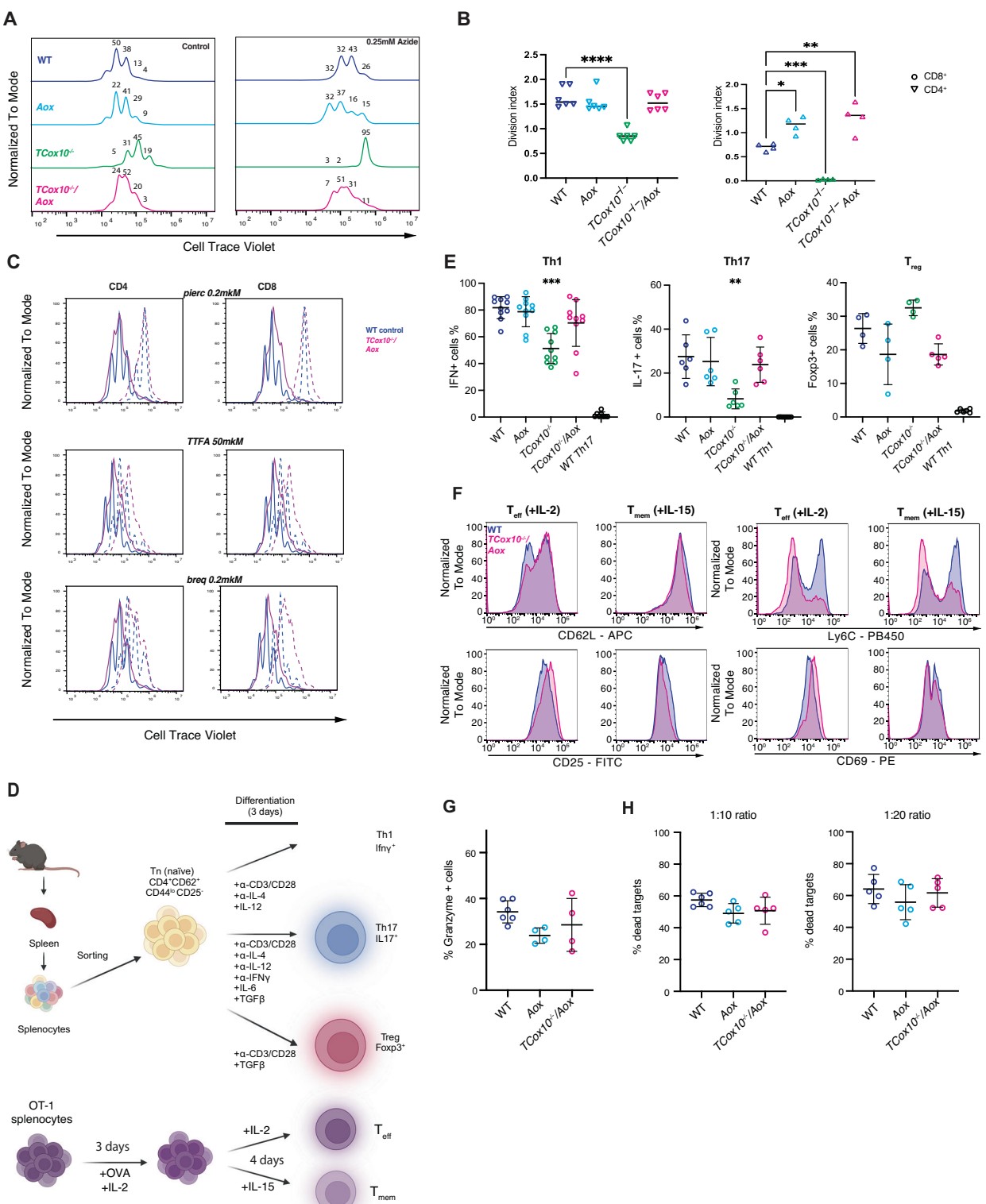

**Fig. 6 | AOX-MR sustains T cell function in vitro. A** Proliferation analysis (Cell Trace Violet) of T cells without and following treatment with 0.25 mM sodium azide. Freshly isolated splenic pan T cells were loaded with Cell Trace Violet and stimulated for 3 days as above with or without sodium azide (0.25 mM). **B** Division index with addition of sodium azide in all genotypes for CD4+ and CD8+ T cells (*N* = 4–6/condition). **C** Proliferation of WT (blue) and *TCox10−/−/Aox* (red) CD4+ and CD8+ cell populations in control (solid line) and with addition of inhibitors (dashed line): piericidin (0.2 μM, complex I), TTFA (50 μM, complex II), and brequinar (0.2 μM, DHODH). **D** Schematic of differentiation for Th1, Th17, and Treg cells. Lower panel is schematic of OT-1 splenocyte differentiation into T_eff and T_mem subtypes. Created in BioRender. Mcguire, P. (2025) https://BioRender.com/

970clph. **E** Percent IFNγ+ (Th1), IL17+ (Th17), and Foxp3+ (Treg) positive cells after differentiation (*N* = 4–10/condition). **F** Surface expression of T_eff and T_mem activation markers between IL-2 or IL-15 differentiated WT and *TCox10−/−/Aox* CD8+ T cells. **G** Percent granzyme positive T cells across genotypes (*N* = 4–6/condition). **H** Percent dead targets across genotypes with 1:10 and 1:20 effector/target ratio (*N* = 5–6/condition). Data are representative of at least three independent experiments and number indicate percentage of cells among groups. indicate mean and standard deviation. * *p* < 0.05, ** *p* < 0.01, *** *p* < 0.001 by one-way ANOVA and post-hoc Dunnett test against WT unless otherwise specified below. Negative controls are not included in statistical calculations.

accompanied by normalization of activation phenotype. Flow cytometry analysis revealed that activated $TCox10^{-/-}$ T cells displayed cell surface markers indicative of an abnormal, heightened activation state (CD44, CD69), while $TCox10^{-/-}/Aox$ cells more closely aligned to WT (Supplementary Fig. 4).

With the restoration of proliferation and normalization of activation markers, we next evaluated the role of AOX-MR in T cell differentiation. We began by assessing CD4+ helper T cells (Th), as they encompass diverse functional subsets with distinct metabolic requirements. $TCox10^{-/-}/Aox$ Th cells were generated in vitro using standardized differentiation protocols (Fig. 6D). We found that $TCox10^{-/-}/Aox$ T cells regained the capacity to differentiate into effector subsets, particularly Th1 and Th17 cells, approximating the patterns observed in WT (Supplementary Fig. 5A and Fig. 6E). In contrast, regulatory T cells (Foxp3+, Treg) were reduced in AOX-expressing cells, albeit not significantly, suggesting a more nuanced or subset-specific relationship between AOX-MR and lineage specification. We next evaluated T cell effector ($T_{eff}$) and memory ($T_{mem}$) differentiation in vitro (Fig. 6D), essential processes for sustained immune protection. In $TCox10^{-/-}$ T cells, memory and effector cell differentiation were previously unachievable due to overwhelming apoptosis and death. The re-introduction of AOX-MR facilitated the generation of both $T_{eff}$ and $T_{mem}$. These cells exhibited improvements in the expression of phenotypic markers of differentiation for their respective cell types, albeit incompletely (Fig. 6F). To assess the functionality of $T_{eff}$ cells, we stained for the expression of granzyme, an essential molecule for cytolytic activity. Granzyme levels in $TCox10^{-/-}/Aox$ matched those in WT and $Aox$ controls, and their killing activity was similarly robust, confirming that the cytotoxic capabilities were fully present (Fig. 6G, H). Given the lack of a robust in vitro assay for memory function, we performed RNAseq analysis on $TCox10^{-/-}/Aox$ differentiated memory T cells (Supplementary Fig. 5B and Supplementary Data 4). Despite significant differences in their transcriptional profiles, the expression of core genes involved in effector memory ($T_{em}$) and central memory ($T_{cm}$) differentiation[28] was largely similar between $TCox10^{-/-}/Aox$ and WT cells, with the notable exception of Eomes. This indicates that while $TCox10^{-/-}/Aox$ memory T cells exhibit broad transcriptional changes, essential pathways for memory differentiation mostly remain intact.

### AOX-MR sustains T cell function in vivo

T cells orchestrate and execute the immune response through cytokine production or direct cellular interactions, serving as both regulators and effectors[29]. A core function of T cells is maintaining immune memory, ensuring a rapid and effective response to previously encountered antigens[30,31]. To evaluate how AOX-MR may support these diverse abilities, we studied $TCox10^{-/-}/Aox$ T cell development and function in vivo. As a secondary lymphoid organ, the spleen serves as a microenvironment for immune interactions, providing a specialized niche where immune cells such as T cells, B cells, and macrophages coordinate responses. In the spleens of both $TCox10^{-/-}$ and $TCox10^{-/-}/Aox$ mice, despite having similar numbers of splenocytes, we observed a tendency for elevated frequencies of B cells and macrophages, signifying imbalances between splenic resident cells (Fig. 7A, B). Despite AOX-MR, decreased quantities of both CD4+ and CD8+ T cells persisted in the spleens of $TCox10^{-/-}/Aox$ mice (Fig. 7C). This reduction signals that AOX-MR, while mitigating certain defects in T cell function, is insufficient to normalize homeostatic proliferation in T cells. With these numeric perturbations in splenic populations, we next measured the ability of T cells to support B cell responses to T-dependent antigens through immunization with 2,4,6-Trinitrophenyl-Chicken Gamma Globulin (TNP-CGG). The results were promising: unlike $TCox10^{-/-}$, $TCox10^{-/-}/Aox$ mice mounted effective primary (2 weeks) and secondary (5 weeks) B cell responses, illustrating that AOX-MR enhances this supportive role of T cells (Fig. 7D).

Given that AOX-MR restored T cell activation and differentiation in vitro, we next tested whether it could support proliferation in vivo. We evaluated proliferative capacity using an adoptive transfer model in OVA-specific T cells (Fig. 7E, F). Following antigen encounter, $TCox10^{-/-}$ T cells showed impaired proliferation, as evidenced by reduced CellTrace Violet (CTV) dilution. In contrast, $TCox10^{-/-}/Aox$ cells exhibited improved CTV dilution, though not to the extent of WT, indicating a partial rescue of proliferative fitness in vivo.

Mitochondria are integral to the development of T cell memory by providing the necessary metabolic and signaling pathways that support their long-term survival and rapid response capabilities[32]. To assess the requirement of AOX-MR for development of memory T cells, we conducted an in vivo challenge with influenza virus (Fig. 7G). Mice were first immunized with influenza A/X31 (X31, H3N2) followed with influenza A/PR/8 (PR8, H1N1) challenge at 5 weeks. Both immunization and challenge were conducted using inhalation. Our experimental design, based on a switch in viral isotypes (X-31 → PR8), eliminates the memory humoral responses, allowing us to focus on T cells. To demonstrate the generation of influenza-specific memory, we stained T cells with tetramers against two T cell antigenic determinants, the nucleoprotein ($NP_{366-374}$, H2Db) and the acid polymerase ($PA_{224-233}$, H2Db)[33,34]. The primary CD8+ T cell response to both strains is dominated by naïve T cell recognition of both determinants. However, the $NP_{366-374}$ response dominates the secondary response in X-31 → PR8 isotype switch challenge. $TCox10^{-/-}$ mice showed a somewhat limited ability to generate memory cells, while $TCox10^{-/-}/Aox$ mice generated memory T cells at levels comparable to WT (Fig. 7H). The most direct evidence of functional recovery came from viral load assessments, where $TCox10^{-/-}/Aox$ mice exhibited viral loads not different from WT, lower than those seen in $TCox10^{-/-}$ mice (Fig. 7I). This dramatic reduction in viral load highlights the restored antiviral efficacy of $TCox10^{-/-}/Aox$ T cells.

### Discussion

While glycolysis has been extensively studied during the metabolic reprogramming following T cell activation, the specific role of MR in this phase remains incompletely defined. Yet this window is metabolically decisive, setting the stage for proliferation, differentiation, and survival. To dissect the requirement for MR, we used AOX as a mechanistic tool. This system allowed us to isolate the core respiratory contribution of the ETC and revealed that MR is necessary to support key metabolic and functional outcomes downstream of activation. Mechanistically, our findings show that continuous electron input into the CoQ pool is required to sustain MR-driven outputs. Complex I contributes electron flux, supports mitochondrial membrane potential, and enables ATP synthesis. Complex II and DHODH also feed electrons into the CoQ pool but do not contribute to proton pumping. These inputs highlight the central role of the ETC in shaping T cell responses. Together, these findings recenter MR as a foundational driver of T cell fate decisions, with implications for how bioenergetic and redox cues orchestrate immune function.

Beyond its metabolic role, MR disruption also triggered activation of the extrinsic apoptotic pathway via Fas/FasL signaling. This was not merely a downstream consequence of impaired proliferation but appears to be a primary effect of disrupted MR. While mitochondria are classically associated with intrinsic, cytochrome c-mediated apoptosis, our data reveal that defects in COX-dependent respiration can also provoke extrinsic apoptotic signaling. Mitochondrial signaling appears capable of regulating Fas/FasL expression itself. In high-glucose stressed retinal epithelial cells, mitochondrial dysfunction led to upregulation of Fas/FasL expression via the JAK/STAT and SOCS1 pathways, while knockdown of Fas or SOCS1 preserved mitochondrial integrity and reduced apoptosis[35]. Similarly, cadmium-induced mitochondrial stress in neurons activated Fas/FasL signaling, with inhibition of ROS or caspase-8 preventing both mitochondrial collapse and

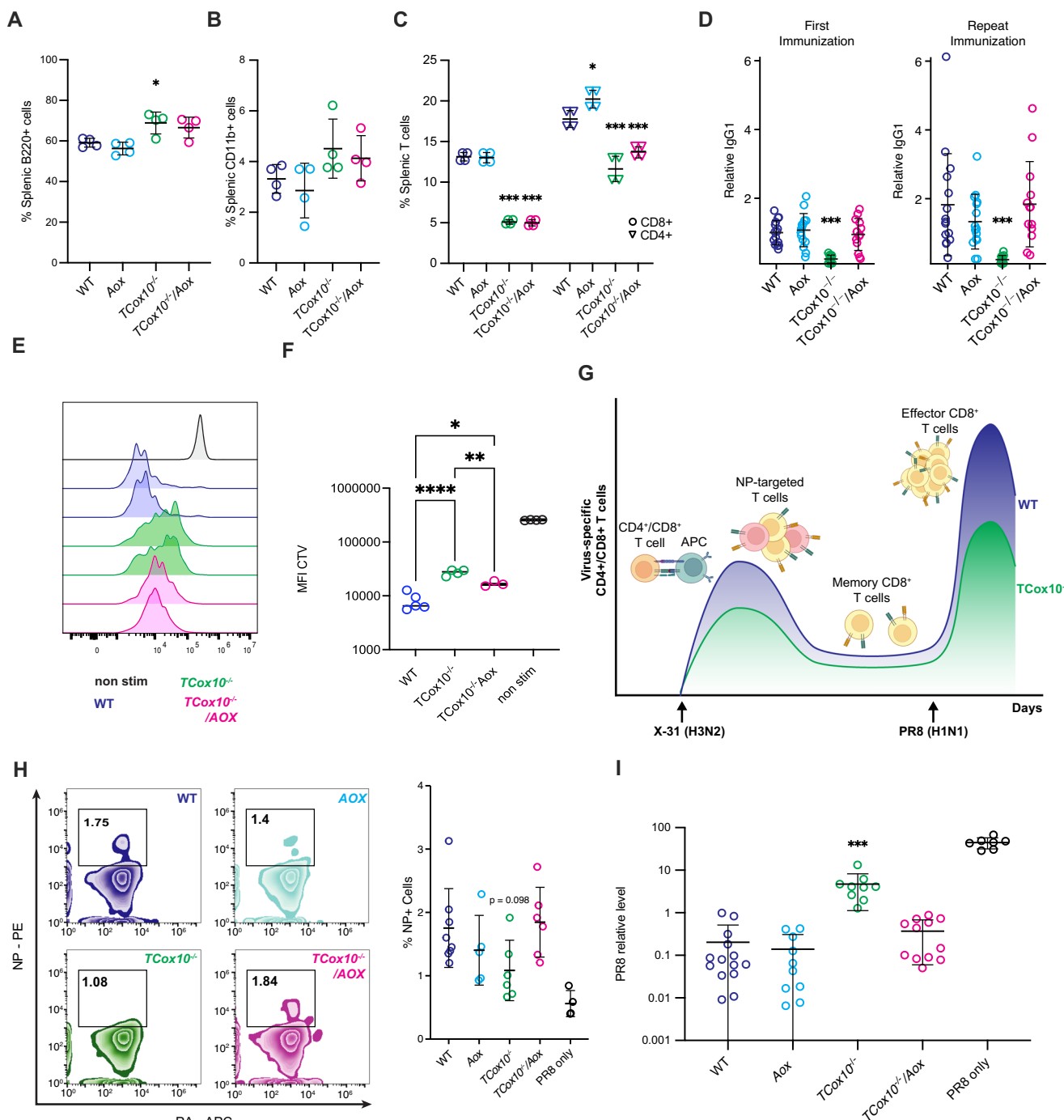

**Fig. 7 | AOX-MR sustains T cell function in vivo.** Splenocytes were collected from WT (mean = 86 × 10⁶ cells/spleen), *Aox* (mean = 86 × 10⁶ cells/spleen), *TCox10⁻/⁻* (91 × 10⁶ cells/spleen), and *TCox10⁻/⁻/Aox* (79 × 10⁶ cells/spleen), *p* = n.s. for all versus WT. **A** Percentage of B220⁺ B cells isolated from spleen of mice of different genotypes. (*N* = 4/condition). **B** Percentage of CD11b⁺ cells isolated from spleen *N* = 4/condition). **C** Percentage of CD8⁺ or CD4⁺ T cells isolated from spleen (*N* = 4/ condition). **D** Relative T cell-dependent antigens (IgG1) detected after first immunization at 2 weeks (left) and reimmunization 1 week after secondary immunization (right) (*N* = 9–11/condition). Control wells are coated with antigen, without serum. Relative values represent absorbance from antigen-coated wells minus background absorbance (i.e., control wells). **E** In vivo proliferation of CD8⁺ cells from different genotypes isolated from CD45.2 OT1 positive mice and adoptively transferred to CD45.1⁺ C57Bl6 (5 × 10⁵ cells/mouse). Mice were injected with ovalbumin to activate in vivo proliferation. Proliferation was tracked via CellTrace Violet dilution and gating was performed on CD45.2⁺ CD8⁺ T cells. **F** Mean fluorescent intensity (MFI) of Cell Trace Violet in CD45.2 gated CD8⁺ cells (*N* = 5–7/condition). **G** Schematic of

deficient development of effector CD8⁺ T cells (Teff) in *TCox10⁻/⁻* mice following viral antigen rechallenge. Mice are initially exposed to X-31, a mouse-adapted non-lethal H3N2 influenza virus, which stimulates the development of T cells targeted to X-31 peptides, including nucleoprotein (NP). Antigen-specific memory CD8⁺ T cells (Tmem) persist following the initial infection. Mice are then exposed to PR8, a mouse-adapted lethal H1N1 influenza virus, which shares the NP antigen with X-31. Created in BioRender. Mcguire, P. (2025) https://BioRender.com/x5ubbtd. **H** Flow cytometry staining of NP⁺ and PA⁺ tetramer positive CD8⁺cells in X31/PR8 exposed mice (*N* = 5–8/condition). Quantification of NP⁺ CD8⁺ T cells following X31/PR8-exposure (right). **I** PR8 relative viral load in X31/PR8-exposed mice measured by qPCR in lung tissue (*N* = 9–14/condition). Data are representative of at least three independent experiments and indicate mean and standard deviation. * *p* < 0.05, ** *p* < 0.01, *** *p* < 0.001 by one-way ANOVA and post-hoc Dunnett test against WT unless otherwise specified below. Controls are not included in statistical calculations.

Fas activation[36]. mtROS also play a regulatory role in immune cells; in memory-like CD4[+] T cells, loss of Fas results in elevated mtROS and IFN-γ, while FasL treatment suppresses these mitochondrial signals, demonstrating a feedback loop between Fas signaling and mitochondrial oxidative stress[37]. Together, these findings expand the view of mitochondrial apoptosis due to COX deficiency in T cells, showing that impaired MR not only triggers intrinsic death pathways but can also initiate or amplify extrinsic apoptosis through Fas/FasL signaling. These death signals underscore how impaired MR reverberates across multiple T cell programs, from survival to bioenergetic wiring, prompting a closer look at how specific mitochondrial inputs shape this dependency.

T cell immunometabolism has long drawn on paradigms from cancer biology, particularly the reliance on aerobic glycolysis to fuel proliferation[1]. Yet this framework often fails to capture critical distinctions in how T cells use mitochondrial pathways. For example, Complex I deficiency has divergent effects across cancer models; impairing proliferation in some by reducing ATP and elevating oxidative stress, while in others enhancing tumor growth by disrupting redox balance and activating proliferative pathways like Akt/mTOR[38,39]. In contrast, Complex I inhibition in T cells using rotenone suppresses CD8[+] T cell proliferation, cytokine secretion, and cytotoxicity, highlighting a non-redundant requirement for mitochondrial input[40]. Similarly, Complex II (succinate dehydrogenase) loss supports transformation in some cancers but in T cells leads to impaired survival and proliferation due to disrupted TCA cycling and pyrimidine synthesis[41]. DHODH adds further complexity. Though essential for de novo pyrimidine biosynthesis, its disruption primarily affects nucleotide pools and often spares core respiratory function[42–45]. In $TCox10^{-/-}/AOX$ T cells, DHODH inhibition modestly impaired membrane potential, while mitochondrial ATP production was largely preserved. This suggests that DHODH is not essential for maintaining core mitochondrial bioenergetics, although it remains required for proliferation due to its role in pyrimidine biosynthesis. Indeed, inhibition of Complex I, Complex II, or DHODH impaired proliferation across genotypes, indicating that while their functions differ, each contributes critically, whether by driving respiration, sustaining redox balance, or supporting biosynthesis. These results caution against assuming that ETC roles are universally conserved across cell types. Even when AOX preserves downstream electron flow, loss of upstream inputs can disrupt the integrated circuit. This underscores the importance of mechanistically defining electron transport requirements in a cell-type-specific context to understand T cell immunometabolism.

In addition to these bioenergetic and biosynthetic requirements, our RNAseq analysis showed induction of ATF4 target genes, consistent with activation of the mitochondrial integrated stress response. The ISR is known to inhibit global protein translation while selectively allowing stress-responsive programs[46], and thus its engagement could contribute to the impaired proliferation we observed in MR-deficient T cells. This raises the possibility that ISR signaling, alongside impaired respiration, acts as an additional layer constraining T cell expansion.

In $TCox10^{-/-}$ T cells, we observed increased ROS, stabilization of HIF1α, and elevated glycolysis, consistent with classical redox-driven metabolic reprogramming. Although mtROS are classically linked to HIF1α stabilization and increased glycolysis under hypoxic conditions[47], our findings suggest that additional mitochondrial signals may regulate glycolysis under normoxic conditions. In $TCox10^{-/-}/Aox$ T cells, mitochondrial superoxide levels are low, Complex III is bypassed, and glycolysis remains elevated despite restored MR. One possibility is that reduced redox signaling contributes to this phenotype. ROS follow a Goldilocks principle in which insufficient ROS fail to engage signaling and excessive ROS cause damage[48]. Within an optimal window, controlled ROS act as second messengers that modulate metabolism. Prior studies show that ROS can oxidatively inactivate glycolytic enzymes such as GAPDH and PKM2[49,50], activate AMPK and

suppress mTORC1[51], and influence transcription through NFAT and NF-κB[4,52]. Other mitochondrial outputs may also influence glycolysis, including NAD[+]/NADH balance, calcium handling, membrane potential, and metabolite export such as citrate, succinate, and α-ketoglutarate. Many of these measures, and TCA cycle activity, were normalized by AOX in $TCox10^{-/-}/Aox$ T cells, and our RNAseq analysis did not show a broad transcriptional induction of glycolytic enzyme genes, which argues against primary transcriptional upregulation. Nevertheless, epigenetically active metabolites can couple mitochondrial metabolism to gene expression. Acetyl-CoA availability regulates histone acetylation and expression of metabolic genes[53,54], and one-carbon metabolism controls cellular S-adenosylmethionine pools and histone and DNA methylation[55,56]. The persistent glycolysis could therefore reflect the absence of specific mitochondrial signals, redox-dependent or metabolite-driven, that remain to be defined.

Together, these findings position MR not simply as an ATP source but as a central coordinator of T cell metabolic and signaling programs. By using AOX as a mechanistic probe, we isolated the respiratory contribution of the ETC and showed how defects in MR reverberate across proliferation, survival, apoptosis, and glycolytic control. Our data also highlight that the consequences of impaired MR encompass integrated stress responses, altered redox signaling, and extrinsic death pathways. These insights expand the framework of T cell immunometabolism, underscoring the need to define how discrete mitochondrial inputs shape immune cell fate.

## Methods
### Mouse lines
C57Bl/6J (WT, B6) (strain #000664), Tg(Cd4-cre)1Cwi/BfluJ (Cd4-cre) (strain #017336), and C57BL/6-Tg(TcraTcrb)1100Mjb/J (OT-1) (strain #003831) mice were purchased from Jax. B6.129X1-Cox10tm1Ctm/J (Cox10fl/fl) were a kind gift from C. Morales and F.Diaz (University of Miami). *Rosa26*-targeted *Ciona intestinalis* alternative oxidase expressing (Aox) mice were a generous gift from Drs. Marten Szibor and Howard T. Jacobs[12]. Mice were housed in a specific pathogen-free facility, maintained at 20–24 °C with 40–60% humidity, and kept on a 12-h light/dark cycle. Animals had ad libitum access to standard chow and water. All procedures involving animals were in compliance with the Animal Care and Use Committee of the National Human Genome Research Center under an established protocol (G-11-3).

### Cell culture
Bulk T cell preparations were isolated from spleen with Pan-T cell Separation kit (Miltenyi Biotec) and stimulated ex vivo for 24 or 72 h with plate bound anti CD3 and anti-CD28 (BioXCell) in complete RPMI medium supplemented with 10% fetal bovine serum (FBS), 2 mM L-glutamine, 100 U/ml penicillin, 1 mM sodium pyruvate, and 50 μM β-mercaptoethanol before collection. T cells were stimulated with plate-bound anti-CD3 (5 μg/ml) and anti-CD28 (0.5 μg/ml). Cells were treated with inhibitors sodium azide with varying concentrations as described in the results. Piericidin (cat#15379), 2-Thenoyltrifluoroacetone (TTFA, cat#15517), brequinar (cat 24445) were purchased from Cayman Chemicals, MitoPQ (#SML3152) from Millipore Sigma, LDHA inhibitor GSK 2837808A (cat# 5189) from Tocris Biosciences.

### Stable isotopes
Mouse T cells were stimulated for 24 h with plate-bound anti-CD3/CD28. Labeling experiments were performed in complete RPMI medium supplemented with 10% FBS (non-dialyzed)[57,58]. All labeling experiments were performed with 1 million cells/mL cultured in RPMI containing 11 mM glucose and 2 mM glutamine, with one nutrient or the other replaced by a uniformly [13]C-labeled analog (i.e., [U – [13]C] glucose or [U – [13]C] glutamine; Cambridge Isotope Laboratories). Cells were rinsed in phosphate-buffered saline, then replenished with labeling medium at time 0. Culture proceeded for 24 h, then the cells

were briefly rinsed in cold saline, pelleted, and lysed in cold 50% methanol. The lysates were subjected to at least three freeze-thaw cycles, then centrifuged to remove debris. The supernatants were evaporated to dryness, methoximated, and derivatized by tert-butyl dimethylsilylation. One μL of the derivatized material was injected onto an Agilent 6970 gas chromatograph equipped with a fused silica capillary GC column (30 m length, 0.25 mm diameter) and networked to either an Agilent 5973 or a 5975 Mass Selective Detector (UTSW Metabolomics Core Facility, Dallas, TX). Retention times of all metabolites of interest were validated using pure standards. The measured distribution of mass isotopomers was corrected for natural abundance of $^{13}C$[59].

## RNAseq

T cells ($10^6$ cells) were lysed in TRIzol (Thermo Fisher Scientific). Total RNA was isolated and purified with RNeasy kit (Qiagen) according to manufacturer's protocol. RNAseq was performed by an outside commercial laboratory (Novogene, Sacramento, CA). Messenger RNA was purified using poly-T oligo-attached magnetic beads. First-strand cDNA was synthesized with random hexamer primers, followed by second-strand cDNA synthesis using dTTP. Libraries then underwent end repair, A-tailing, adapter ligation, size selection, amplification, and purification. Sequencing was performed on the Illumina NovaSeq 6000 with 150 bp paired-end reads. Reads were aligned to reference genome mm10 with Hisat2 v2.0.5, and raw read counts were determined using FeatureCounts v1.5.0-p3. Raw count normalization and differential expression analysis was performed using DESeq2 1.42.0[60]. Hierarchical clustering was performed for outlier detection (Supplementary Fig. 6A). Volcano plots were prepared using EnhancedVolcano 1.20 with apeglm fold change shrinkage[61]. For subsequent analysis, genes were considered significant if padj < 0.05.

## GSEA

Gene sets were ranked by t-stat determined by DESeq2 prior to enrichment analysis. GSEA and visualization was performed with clusterProfiler 4.11.0[62]. MitoCarta pathways were derived from a subset of mouse MitoCarta3.0 MitoPathways[63], with the addition of GO:0006098 (pentose-phosphate shunt) and GO:0061621 (canonical glycolysis). Visualization of KEGG pathway "apoptosis" (mmu04210) was performed with pathview 1.42.0 and KEGGREST 1.42.0.

## Transmission electron microscopy

After 24 h of stimulation T cells were fixed for 48 h at 4 °C in 2% glutaraldehyde and 1% paraformaldehyde in 0.1 M cacodylate buffer (pH 7.4) and washed with cacodylate buffer three times. The T cells were fixed with 1% $OsO_4$ for two hours, and washed again with 0.1 M cacodylatebuffer three times. The pellets were subsequently serially dehydrated in ethanol and propylene oxide and embedded in EMBed 812 resin (Electron Microscopy Sciences, Hatfield, PA, USA). Thin sections, approx. 80 nm, were obtained by utilizing the Leica ultracut-UCT ultramicrotome (Leica, Deerfield, IL, USA) and placed onto 300 mesh copper grids and stained UranyLess solution (Electron Microscopy Sciences, Hatfield, PA, USA), methanol, and then with lead citrate. The grids were viewed in the JEM-1200EXII electron microscope (JEOL Ltd, Tokyo, Japan) at 80 kV and images were recorded on the XR611M, mid-mounted, 10.5 Mpixel, CCD camera (Advanced Microscopy Techniques Corp, Danvers, MA, USA).

## Mitochondrial morphology quantification

To analyze mitochondrial morphology, transmission electron microscopy (TEM) images were processed using Fiji (ImageJ) software and analysis was done using a previously published method[64]. Images were collected from at least 3–4 independent experiments. For the analysis, each image was exported with a scale bar, which was used to calibrate the measurements in Fiji so that all images had uniform units. The entire cell in each image was first outlined using the freehand selection tool and saved to the ROI Manager. Next, each individual mitochondrion within the cell was carefully traced using the same tool and added to the ROI Manager. After all mitochondria were traced, the "Measure" function was used to obtain measurement of each shape of mitochondrion.

## Influenza infection

Mouse adapted human influenza virus A/PR/8/34 (PR8) and A/X/31 (X31) and were used for infection. Mice ($N = 10$/genotype) were exposed to aerosolized (Glas-Col) 500 $TCID_{50}$ of X31 or PR8 in 7 mL of saline. Details and time points of infection are outlined in the text. Expression of viral hemagglutinin (HA) in the lungs of infected mice was determined by real-time PCR (primers, Thermo-Fisher).

## Cell transfer

CD8 cells from OT1 mice were isolated by magnetic beads separation and (CD45.2, $5 \times 10^5$ cells) were injected retroorbitally into 8–12 week old B6 mice (CD45.1). Mice were injected with 50 μg/mice albumin from chicken egg (A5503, Millipore Sigma). Mice were then euthanized at time points as defined in the results.

## Flow cytometry

Flow cytometric analysis were performed using a CytoFLEX LX cytometer (Beckman Coulter) equipped with 405, 488, 561, and 638 lasers and analyzed using FlowJo software (Tree Star). Anti-CD4, CD8, CD44, B220, CD11b, CD25, CD69, CD247, IgM, CXCR5, PD-1, Foxp3, annexinV antibody were purchased from BD Biosciences or Thermo Fisher Scientific. LIVE/DEAD fixable aqua(L34957, Thermo Fisher) and ViaKrome 808 fixable (C36628, Beckman Coulter) dies were used to determine viability. Labeled tetramers (NIH tetramer core facility) were used to identify virus-specific T cells. MitoSOX red (5 μM), Total Reactive Oxygen Species (1×, 100 μL) assay kit, and TMRE (100 nM) (Thermo Fisher), MitoPY1 kit (5 μM, Tocris Bioscience), ATP-Red (20 μM, Millipore Sigma), were used according to manufacturer instructions. Apoptosis was measured by Annexin V staining (ebioscience). Substrate cleavage by caspases were measured with caspase substrates PhiPhilus-G1D2, CaspaLux9-M2D2, CaspaLux8-L1D2 (OncoImmunin Inc.) according to the manufacturer instructions. Cells were loaded with 5 μM CTV (Thermo Fisher Scientific) and proliferation were estimated on day 3 by FACS. Gating strategy for CD4+ and CD8+ T cells is shown in Supplementary Fig. 6B.

## Real-time PCR

RNA was extracted from the tissues using Pure Link RNA mini kit (Thermo Fisher Scientific) and was reverse transcribed to cDNA (iScript, Bio-Rad) according to the manufacturer's instructions. Reactions were cycled and quantitated with an CFX96 Real Time BioRad PCR System (BioRad Laboratories) (Primers, Thermo-Fisher).

## In vitro differentiation and killing assay

Naive (CD4+ (PerCP/Cy5.5, clone RM4−5) CD44low (APC, clone IM7) CD62Lhi (eFluor 450, clone MEL-14) CD25neg (PE, clone PC61.5) T cells) were purified by cell sorting. Purity was greater than 99%. Sorted naive CD4+ T cells ($2 \times 10^5$) were co-cultured at a ratio of 1:5 with mitomycin-treated T-depleted splenocytes as APCs in 48-well plates under various differentiation conditions for 3 days. Th1 conditions included 40 ng/ml IL-12 and anti-IL-4.Th17 used 20 ng/ml of IL-6, 5 ng/ml of TGF-β1, anti-IL-4, anti-IFN-γ, and anti-IL-12. Treg used 100 U/ml hIL-2, 5 ng/ml TGF-β1 with 1 μg anti-CD3, and 10 μg/ml of each anti-IL-4, anti-IFN-γ, and anti-IL-12 antibodies. For effector and memory T cell preparation, splenocytes from mice crossed with OT1 transgenic mice were isolated, stimulated for three days with OVA(257−264) peptide (Anaspec), followed by wash and incubation for four more days with IL-2 or IL-15 (10 ng/ml), (R&D systems) before collection as indicated in the text.

Pan T cells, CD8[+] T or CD4[+] cells were enriched using isolation kits (Miltenyi Biotec). Purity of T cells was >95% in all cases. Antibodies were purchased from BioXcell unless otherwise indicated. For cytotoxicity assay was splenocytes from mice crossed with OT1 transgenic mice were isolated, stimulated for three days with OVA(257–264) peptide (Anaspec), followed by wash and incubation for four more days with IL-2 (10 ng/ml). EL-4 cells as targets were loaded with CTV and incubated for one hour with OVA peptide (257–264), then washed and co-incubated with cytotoxic T cells in different ratio of effecter/target for 4 h. Amount of killed cells were determined by FACS.

## OCR and ECAR measurement

OCR and ECAR) were measured using a Seahorse XF$^e$96 analyzer (Agilent Technologies). Pan T cells, CD4[+] or CD8[+] T cells from mice activated for 24 h with anti CD3 and anti-CD28 were attached with Cell-Tak (Corning) according to manufacturer's instructions at concentration 0.2 million cells/well in Seahorse Base medium supplemented with 2 mM L-glutamine, 1 mM sodium pyruvate with addition of 25 mM glucose. OCR and ECAR were determined according to the manufacturer's standard protocol in response to 1 μM oligomycin, 1 μM FCCP, 100 nM rotenone + 1 μM antimycin, and 50 mM 2-deoxy-glucose (2-DG). OCR and ECAR were calculated and recorded by the Seahorse XF$^e$96 software. Complex IV activity (COX) was measured according to published methods[65] using tetramethyl-p-phenylenediamine (TMPD) as an electron donor that is specific for complex IV with OCR as the readout.

## Immunization and serum analysis

Mice were immunized with 50 μg of TNP-CGG in Imject Alum (Pierce Chemical) and re-immunized with TNP-CGG in 28 days. Sera were tested by ELISA for TNP reactivity. Briefly, plates were coated with TNP-BSA or (10 μg/ml; Biosearch Technologies), and bound immunoglobulins were detected by alkaline phosphatase-conjugated detection antibodies to specific mouse isotypes (Southern Biotech).

## Lactate measurement

Production of lactate in the medium of pan T cells at a density of $2 \times 10^6$ cells/ml stimulated in complete RPMI media with anti-CD3 and anti-CD28 for 24 h was measured amperometrically using a YSI 2950 Analyzer (Xylem) in triplicate and averaged. Changes in metabolite concentration were determined by measuring metabolite concentrations in culture media incubated in the absence of cells and subtracting these values from those measured in culture media supernatants obtained from wells incubated with cells. YSI metabolite measurements were made in the Clinical Cancer Metabolism Facility of the Center for Cancer Research, National Cancer Institute.

## Statistical analysis and visualization

Statistical significance was determined using GraphPad Prism v10 or R 4.3.2. All flow cytometry data were analyzed with FlowJo v10. Statistical tests used and n for each experiment are indicated in each figure legend. Sample sizes used are similar to our previous publication[6]. Experiments and data analysis were not performed blinded to genotype. Data distribution was assumed to be normal. Data was plotted and visualized with GraphPad Prism v10, or R packages clusterProfiler 4.11.0, ggplot2 3.5, pheatmap 1.0.12, or ComplexHeatmap 2.18.0[66]. Figures were prepared using BioRender and Adobe Illustrator.

## Reporting summary

Further information on research design is available in the Nature Portfolio Reporting Summary linked to this article.

## Data availability

Processed data used in the analysis are available via the Source Data File for purposes of reproducing or extending the analysis. RNAseq data is available at GEO under accession GSE269797. All other data are available in the article and its Supplementary files or from the corresponding author upon request. Source data are provided with this paper.

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

## Acknowledgements
We would like to acknowledge Ms. Stacie Anderson and Ms. Martha Kirby for their assistance with cell sorting and flow cytometry (NHGRI Flow Cytometry Core) and Dr. Crooks from Clinical Cancer Metabolism Facility (NCI) for help with lactate measurement. We would also like to thank Dr. Howard T. Jacobs (Tampere University) for the Mit-AOX mouse and Dr. Afshin Beheshti for advice on RNAseq analyses. Funding for this project was provided by the National Institutes of Health grant ZIA HG200381-12 (P.J.M.).

## Author contributions
Conceptualization: T.T., P.J.M. Methodology: T.T., E.W., A.F., B.S., J.M., M.S.Z. Investigation: T.T., E.W., A.F., B.S., J.M., C.K. Visualization: T.T., E.W. Funding acquisition: P.J.M. Project administration: T.T., E.W., P.J.M. Supervision: P.J.M. Writing – original draft: T.T., E.W., P.J.M. Writing – review & editing: T.T., E.W., A.F., B.S., J.M., M.S.Z., P.J.M.

## Funding

## Competing interests
The authors declare the following competing interests: M.S.Z. is a co-founder of a start-up company founded to develop therapeutics based on AOX. The remaining authors declare no competing interests.
