## [Transparent Peer Review file · Nature Communications]

Cytochrome c oxidase dependent respiration is essential for T cell activation, proliferation and memory formation

Corresponding Author: Dr Peter McGuire

Version 0:

Reviewer comments:

Reviewer #1

(Remarks to the Author)

The authors have previously shown in a Cell Metabolism paper (2017; PMID: 28591633) that Cytochrome c oxidase (by analyzing conditional Cox10 knockout mice) is a metabolic checkpoint for cell fate decisions following T cell activation, with heterogeneous effects in T cell proliferation, differentiation and survival of activated T cells. These data indicated several roles for the COX complex in T cells. In the current study, Tarasenko et al investigate how much of the defects observed are due to role of COX-mediated mitochondrial respiration or whether there are other functions of COX not related to MR. In order to do so, the authors crossed conditional Cox10^{f/f} mice (T cell-specific deletion with Cd4-Cre) with mice expressing a related ubiquinol oxidase (AOX) (from Ciona intestinal) from the Rosa26 locus. In this model, T cells (as well as all other cells) express AOX instead of Cox10. The authors then analyzed the metabolic state as well as various immunological function of the reconstituted T cells. By using this approach, and by employing a variety of assays the authors demonstrate that AOX expression restored MR as well as several other aspects of mitochondrial function (reduction of oxidative stress to WT levels, improvement of ATP production and cellular redox balance as well as TCA cycle). This correlated with a rescue of many functional impairments of Cox10-deficient T cells including cell viability, proliferation and differentiation into various Th subsets as well as CD8 effector cells). Based on these data, the authors conclude that the defects observed in the absence of a functional COX complex are due to the impaired MR and therefore that most of the other defects observed upon inactivation of the COX complex in T cells are secondary to the impaired MR. Overall, I think this is a nice experimental strategy and study to dissect the role of COX in more details.

I have a few comments concerning the way some of the data presented data.

(1) In their Cell Metabolism paper, the authors describe that the deletion efficiency of Cox10^{-/-} was approx. 70% in T cells at steady-state in splenic T cells and that this might indicate selective pressure on naive T cells to keep Cox10 (upon activation >95% deletion was reported). What is the deletion efficiency of Cox10 in naive T cells that express AOC?

(2) In figure 4A, the authors should adjust the quadrants. It seems that CD8-negative and CD4-negative cells are included in the CD8⁺ and CD4⁺ quadrants (they should use similar gating as in figure S3B, where the quadrants are set properly). Moreover, the authors should indicate in the legend for figure 4 from which organ the cells were isolated. In addition, based on the information provided, it is not clear whether the cells were activated in vitro or directly analyzed ex vivo. The figure legend title includes the word "proliferation-induced", so it seems that the cells were activated (but it is not stated how they were activated?).

(3) The immunological analysis and data presentation shown in figure 5 should be improved. In figure 5A and 5B, it is not clear which T cell subsets were used. Naive CD4⁺ T cells, naive CD8⁺ or total CD3⁺ T cells (naive?)? In addition, it might be better to indicate the frequency of cells that had 1, 2, 3 rounds of cell division. For figure 5B, the authors should show the frequency of cells that upregulated CD25, CD44 or CD69 instead of showing normalized MFI. In figure 5D, what is the negative control? In addition, showing representative flow cytometry plots would be useful. This also would indicate whether MFI cytokine expression levels are changed or not. In figure 5E the authors should indicate for how long the cells were cultured. In my opinion the culture conditions (3 days IL-15 compared to 3 days IL-2) doesn't really reflect memory T cells- it might be better to rephrase the wording. For the histograms in figure 5F, were the cells gated on CD8⁺ T cells? This should be indicated. In figure 5I, what was the criteria for selecting the genes shown? It might be useful to use a gene set for memory T cells and analyse whether (some of) these genes are expressed at similar levels to make an unbiased statement.

One comment to figure 5C and the figure legend: the authors differentiated naive CD4+ T cells into various Th subsets. In 5C and in the figure legend, the authors name naive CD4+ T cells as Th0. However, the name Th0 is rather used for naive CD4+ T cells that have been activated with anti-CD3/CD28 in the absence of polarizing cytokines, so I recommend to write naive T cells (or Tn) instead of Th0.

(4) The authors crossed the OT-I tg TCR to WT and Tcox10^{-/-}Aox mice. Is there a difference in the composition of peripheral T cell subsets (also distribution naive vs effector cells)? This is important, since OT-1 CD8+ T cells are used in the experiments described in figure 5E and 5F. And how was the cytotoxic assay performed (figure 5H)? This is not clear at all. How was this determined?

(5) In figure 6A-C, the authors show the frequencies of splenic B cells, CD11b+ cells and splenic T cells and observed increased frequencies of B cells and reduced frequencies of T cells. The authors mention in line 261 numeric alterations but don't show data. Is this reflected with changes in T cell numbers (or B cell numbers?)? For the immunization experiments shown in figure 6D, after how many days after the primary and secondary stimulation were the TNP-specific Ab measured - perhaps I overlooked it but I didn't find any information? The Y-axis shows relative values - it is not clear to this reviewer what "relative" refers to (to non-immunised or coated plates with an unrelated Ag?).

(6) In figure 6F, the authors show Ag-specific T cells after infection. The data are not really convincing, based on the graph at the right it seems that there is not really a difference between the various groups of mice so it is not clear whether this is rescued. Do the authors have a non-infected mouse staining control? The frequency of cells within the regions should be indicated.

(7) figure 6H: were the cells gated on CD45.2 cells? The authors mention that they want to test whether the effect was cell-autonomous (T cell autonomous?). Since the authors transferred BM cells (and all express AOX from the Rosa26 locus), it doesn't formally exclude that other hematopoietic cells (that all express AOX) contribute to the rescue of the observed phenotypes. Perhaps I misunderstood the experiment - a better rationale/description from lines 284 on would be helpful to understand this better. Was there a difference in the reconstitution efficiency? And when were the mice infected after BM reconstitution? This is not mentioned.

(8) The information provided in the method section is sometimes very limited. I think it is important to indicate how many cells were stimulated, in which medium and in which volume, cytokine concentration, etc. (e.g. methods in line 368). The authors refer often in the method section to the text in the results, however it would be better to provide details in the methods as well.

minor issues:

line 287: the word "added" should be removed.

line 481: it is not stated what the control coating was.

Reviewer #2

(Remarks to the Author)

Comments:

The manuscript by Tarasenko et al., entitled 'Cytochrome c oxidase dependent respiration is essential for T cell activation, proliferation and memory formation' investigates the role of mitochondrial complex IV in T-cell biology. The authors explore a combination of in vitro and in vivo assays via use of COX-deficient T-cells in combination with an alternative oxidase (AOX) model to bypass the deficiency. The authors introduce the topic well, especially with regards to highlighting the requirement for OxPhos upon T-cell activation – something often overlooked in favour of the Warburg effect. Key results demonstrate that loss of COX leads to heightened apoptosis and metabolic dysregulation with AOX rescuing these phenotypes. The authors demonstrate in an infection model that loss of COX impairs the immunization and challenge response, largely attributed to loss of mitochondrial respiration. The manuscript is novel and timely, delineating the role of CIV in T-cell biology; however, I have a number of concerns that would need to be addressed.

Major

1. TMRE limitations. Accumulations of the probe are known to be affected by mitochondrial mass amongst other factors. If the authors normalised the TMRE to MitoGreen content would the difference still remain in the TCox10^{-/-} group? Could it be that the observed TMRE increase is just a result of an increased mitochondrial content. Controls are also lacking in these experiments (e.g., FCCP). Furthermore, there are multiple issues using TMRE, such as its accumulation is affected by the plasma membrane's potential – have the authors considered whether there would be a difference here. Flow cytometry, as employed, will not tell one where the probe is accumulating (cytosol vs mitochondria). These points may impact on conclusion draw within the paper.

2. Mitochondrial morphology – linking to the point above – are more comprehensive morphological assessment is required. Given the impact on OxPhos, it would be of interest to determine whether the mitochondrial shape, number of cristae etc are impacted by TCox10^{-/-} and TCox10^{-/-}/AOX.

3. 2-NBDG has been demonstrated not to be a reliable indicator of glucose uptake (PMID: 32879737). Please remove all data using 2-NBDG. Alternatively, the authors should use a glucose measurement kit to demonstrate enhanced glucose uptake or use their mass spectrometry data to identify increased intracellular glucose levels if it is within the detection limits.

Minor

1. Introduction: Some key references are missing regarding CIV and T-cell biology (PMIDs: 30626970, 23415911).
2. Given the observed differences in translation in TCox10^{-/-}/AOX cells, are the cells larger via FSC-A? If so, can the authors confirm whether normalisation to cell size has been performed on mitochondrial parameters? Furthermore, the authors could confirm whether translation is globally enhanced by performing a puromycin incorporation assay between the three/four groups.
3. Line 169 – please alter the word uptake to production.
4. Figure 2H – please demonstrate where the injections are added on the plot.
5. What do the corresponding ECAR plots look like for Figure 2H? Do they agree with the stable isotope tracing analysis in Figure 3? One would presume that the TCox10^{-/-} ECAR would be lower.
6. Were the stable isotope tracing experiments performed in the presence of 10% dialysed FBS? This should be stated clearly in the methods rather than reference to previous publications.
7. Do the authors see increased reductive carboxylation in the TCox10^{-/-} T-cells as a potential compensatory mechanism to CIV loss? If so, this should be included and discussed.
8. Figure 5A – only a representative plot is shown, can the authors please add the formal proliferation analysis.
9. Representative plots are missing for Figure 5B.
10. Concentrations of probes lacking in methods, particularly in the flow cytometry section.

Reviewer #3

(Remarks to the Author)

The current study determines the effects of restoring respiratory flux in Cox10 deficient mice by expressing the alternative oxidase (AOX), which enables restoring electron transfer flux initiated at complex I and complex II. Authors find that most of the metabolic and functional changes induced by Cox10 deficiency are reversed by AOX expression. The major concerns stem from a misunderstood gap in knowledge that this paper claims to cover, as well as some of the conclusions that are not supported by the data. More specifically:

- 1) Authors use a statement in the introduction that leads to the interpretation that the major and direct action of complex IV (COX) is to regulate apoptosis, mtDNA maintenance and transcription. It has been widely described for mitochondrial diseases and in many different tissues, including by using AOX, that a defect in electron transport chain activity is the primary event leading to secondary changes in mtDNA maintenance, transcription (i.e. mtISR) and cell death. The text leads to the wrong impression that this is different in lymphocytes and that this study addresses such an inexistent gap. This study only confirms the same connections that have been observed in other tissues, but now in lymphocytes. Authors should tone down claims of novelty regarding that this is the first study testing the connection between respiration, apoptosis and transcription.
- 2) Is the activation of the mitochondrial ISR involved in the defect in proliferation and cell death? How many of the cell death genes identified in the RNAseq converge with the mitochondrial ISR? Mitochondrial respiratory dysfunction, depolarization, decreased ATP/ADP and elevated ROS activate OMA1 protease, which activates HRI to block protein translation and activate ATF4 to induce cell death.
- 3) Authors conclude that reductive stress and ROS production are the major drivers for the defects induced by Cox10 deletion. However, no direct evidence is provided to support this claim, as T-cell death and lack of proliferation could also be a result of decreased mitochondrial ATP synthesis or activation of the mitochondrial ISR. To draw such a conclusion, authors should induce reductive stress in Cox10^{-/-} AOX expressing lymphocytes and determine whether it reverses the benefits induced by AOX expression, via the expression of EcSTH for instance. ATP/ADP should be provided as well to determine whether increased glycolysis to lactate is not effective in restoring this ratio and rather decreased ATP is only reflective a decrease in the nucleotide pool destined for DNA replication.
- 4) Electron microscopy or confocal imaging is needed to establish how much of the changes in the multiple fluorescence probes used by flow cytometry are reporting on membrane potential and ROS rather than just mitochondrial mass. The increases and trend to increase in mitoPY and TMRE fluorescence reported might be just reflecting the increase in mitochondrial mass induced by Cox10, as observed in muscles from patients with MERFF. Thus, when normalized by mitochondrial mass, it is likely that membrane potential and H₂O₂ are decreased in Cox10^{-/-} cells. High-resolution live cell imaging is also needed to confirm that the concentrations of mitosox used for flow cytometry only stain mitochondria and not the nucleus.
- 5) Based on transcriptional analysis and % labeled lactate from glucose, authors claim that glycolysis is decreased in Cox10^{-/-}. However, it is widely known that defects in mitochondrial ATP synthesis are compensated by an increase in glycolysis to lactate, otherwise there is a severe ATP deficiency and NADH accumulation. Thus the decrease in transcript levels might just be reflecting a decrease in proliferative capacity, rather than lower glycolytic flux per se. How do the ECAR in response to oligomycin and lactate excretion rates look like in Cox10^{-/-} cells versus Cox10^{-/-}/Aox cells, basally and when these cells are stimulated to proliferate? Can authors measure extracellular lactate content to determine whether there is increased excretion?

Version 1:

Reviewer comments:

Reviewer #1

(Remarks to the Author)

The authors responded well to my comments.

Reviewer #2

(Remarks to the Author)

The authors have addressed my comments in full. Congratulations on an interesting study.

Reviewer #3

(Remarks to the Author)

The authors addressed all the comments and the main conclusions are now supported by data. I only have minor suggestions for the discussion:

-Authors show that ATF4 targets are engaged and thus the mitoISR. It could be discussed that this ISR activation could be contributing to the decrease in proliferation, as ISR blocks protein translation.

- The mitoPQ result is puzzling. MitoPQ increases the production of superoxide in complex I. This superoxide needs SOD2 activity to generate H₂O₂ and then induce signaling, as proposed by the authors. The expectation is that mitoPQ would further damage Tcox10 \pm mitochondria, as the authors show that complex I activity is needed for AOX to rescue and mitoPQ is expected to damage complex I. A discussion reconciling the established mechanism of mitoPQ action and the effects observed needs to be rewritten, as it seems unlikely that mitoPQ is changing glycolysis by increasing H₂O₂ release from mitochondria.

-I would correct the statements of energy production across the manuscript. Energy is not produced by mitochondria. ATP or heat are produced by mitochondria by transforming the energy contained in nutrients.

REVIEWER COMMENTS

Reviewer #1 T cells (Remarks to the Author):

The authors have previously shown in a Cell Metabolism paper (2017; PMID: 28591633) that Cytochrome c oxidase (by analyzing conditional Cox10 knockout mice) is a metabolic checkpoint for cell fate decisions following T cell activation, with heterogeneous effects in T cell proliferation, differentiation and survival of activated T cells. These data indicated several roles for the COX complex in T cells. In the current study, Tarasenko et al investigate how much of the defects observed are due to role of COX-mediated mitochondrial respiration or whether there are other functions of COX not related to MR. In order to do so, the authors crossed conditional Cox10^{f/f} mice (T cell-specific deletion with Cd4-Cre) with mice expressing a related ubiquinol oxidase (AOX) (from Ciona intestinal) from the Rosa26 locus. In this model, T cells (as well as all other cells) express AOX instead of Cox10. The authors then analyzed the metabolic state as well as various immunological function of the reconstituted T cells. By using this approach, and by employing a variety of assays the authors demonstrate that AOX expression restored MR as well as several other aspects of mitochondrial function (reduction of oxidative stress to WT levels, improvement of ATP production and cellular redox balance as well as TCA cycle). This correlated with a rescue of many functional impairments of Cox10-deficient T cells including cell viability, proliferation and differentiation into various Th subsets as well as CD8 effector cells). Based on these data, the authors conclude that the defects observed in the absence of a functional COX complex are due to the impaired MR and therefore that most of the other defects observed upon inactivation of the COX complex in T cells are secondary to the impaired MR. Overall, I think this is a nice experimental strategy and study to dissect the role of COX in more details.

I have a few comments concerning the way some of the data presented data.

(1) In their Cell Metabolism paper, the authors describe that the deletion efficiency of Cox10^{-/-} was approx. 70% in T cells at steady-state in splenic T cells and that this might indicate selective pressure on naive T cells to keep Cox10 (upon activation >95% deletion was reported). What is the deletion efficiency of Cox10 in naive T cells that express AOX?

Response: *We thank the reviewer for their comments. Deletion efficiency was evaluated in sorted ex vivo CD8⁺ and CD4⁺ TCox10^{-/-} and TCox10^{-/-}/Aox T cells by PCR for gDNA. We found the deletion efficiency was similar (~95%, Fig. 1C) in these unstimulated cells. In our previous paper, we showed Cox10 exon 6 gDNA in PanT cells isolated using magnetic bead separation, which was less pure than cell sorting.*

(2) In figure 4A, the authors should adjust the quadrants. It seems that CD8-negative and CD4-negative cells are included in the CD8⁺ and CD4⁺ quadrants (they should use similar gating as in figure S3B, where the quadrants are set properly). Moreover, the authors should indicate in the legend for figure 4 from which organ the cells were isolated. In addition, based on the information provided, it is not clear whether the cells were activated in vitro or directly analyzed ex vivo. The figure legend title includes the word “proliferation-induced”, so it seems that the cells were activated (but it is not stated how they were activated?).

Response: We thank the reviewer for this careful evaluation. First, we have corrected the gating strategy in the figure (now Fig. 5A). Second, we have updated the figure legend to specify that the T cells were isolated from spleen (Fig. 5). Finally, we have clarified in the figure legend that the cells were stimulated in vitro with plate-bound anti-CD3 and anti-CD28 antibodies prior to analysis.

(3) The immunological analysis and data presentation shown in figure 5 should be improved. In figure 5A and 5B, it is not clear which T cell subsets were used. Naive CD4+ T cells, naive CD8+ or total CD3+ T cells (naive?)? In addition, it might be better to indicate the frequency of cells that had 1, 2, 3 rounds of cell division. For figure 5B, the authors should show the frequency of cells that upregulated CD25, CD44 or CD69 instead of showing normalized MFI. In figure 5D, what is the negative control? In addition, showing representative flow cytometry plots would be useful. This also would indicate whether MFI cytokine expression levels are changed or not. In figure 5E the authors should indicate for how long the cells were cultured. In my opinion the culture conditions (3 days IL-15 compared to 3 days IL-2) doesn't really reflect memory T cells- it might be better to rephrase the wording. For the histograms in figure 5F, were the cells gated on CD8+ T cells? This should be indicated. In figure 5I, what was the criteria for selecting the genes shown? It might be useful to use a gene set for memory T cells and analyse whether (some of) these genes are expressed at similar levels to make an unbiased statement.

One comment to figure 5C and the figure legend: the authors differentiated naive CD4+ T cells into various Th subsets. In 5C and in the figure legend, the authors name naive CD4+ T cells as Th0. However, the name Th0 is rather used for naive CD4+ T cells that have been activated with anti-CD3/CD28 in the absence of polarizing cytokines, so I recommend to write naive T cells (or Tn) instead of Th0.

Response: We thank the reviewer for this comment. Figure 6A (previously 5A) shows proliferation of CD4 positive naïve T cells, CD8 proliferation not presented but showed similar pattern. We also indicate the frequency of cells in each division group. We are not able to show the frequency of the of cells positive for activation markers because all population of cells homogeneously shift during activation (Extended Data Fig. 4). We updated the figure 6E to specify what conditions were used as negative control. (WT Th17 skewed cells were used as negative control for FACS staining for Th1 condition, Th1 skewed used as neg control for Th17 staining and Foxp3 staining). Third, we updated the picture to indicate that total 7 days were needed to develop T memory and T effector cells development (Fig. 6D). We indicated that figure 6F was gated on CD8 cells in the figure legend. The gene sets shown in Figure 5I (now Extended Data Fig. 5B) were selected based on the review by Drs. Susan Kaech and Weiguo titled "Transcriptional control of effector and memory CD8+ T cell differentiation" PMID: 23080391, which highlights key transcriptional regulators associated with memory T cell fate. A reference was also added in the text of the results.

(4) The authors crossed the OT-I tg TCR to WT and Tcox10^{-/-}Aox mice. Is there a difference in the composition of peripheral T cell subsets (also distribution naive vs effector cells)? This is important, since OT-1 CD8+ T cells are used in the experiments described in figure 5E and 5F.

And how was the cytotoxic assay performed (figure 5H)? This is not clear at all. How was this determined?

Response: We thank the reviewer for this comment. Regarding the composition of T cells in OT1 mice, we did not observe a reduced percentage of CD8⁺ T cells in the TCox10^{-/-} or TCox10^{-/-}/Aox groups. In fact, 40–45% of live spleen cells were CD8⁺, while CD4⁺ T cells accounted for only 2–3%. Activation marker analysis showed that the CD8⁺ T cells largely retained a naïve phenotype, with only a slight upregulation of CD25 observed in the Aox and TCox10^{-/-}/Aox groups.

We have also updated the Methods section to clarify the cytotoxicity assay. For the assay shown in now Fig. 6H, effector T cells (as generated in Figure 6D) were co-cultured with CTV-labeled EL-4 target cells loaded with Ova peptides at effector-to-target (E:T) ratios of 1:10 and 1:20. After 4 hours of co-incubation, target cell death was assessed by flow cytometry using viability dyes, and cytotoxicity was calculated based on the percentage of dead target cells. These details are now clearly stated in the Figure 6 legend, Results, and Methods sections.

(5) In figure 6A-C, the authors show the frequencies of splenic B cells, CD11b⁺ cells and splenic T cells and observed increased frequencies of B cells and reduced frequencies of T cells. The authors mention in line 261 numeric alterations but don't show data. Is this reflected with changes in T cell numbers (or B cell numbers)? For the immunization experiments shown in figure 6D, after how many days after the primary and secondary stimulation were the TNP-specific Ab measured - perhaps I overlooked it but I didn't find any information? The Y-axis shows relative values - it is not clear to this reviewer what "relative" refers to (to non-immunised or coated plates with an unrelated Ag?).

Response: We thank the reviewer for these observations. The data referenced in the comment appear in Figure 7A–C, where we show the frequencies of splenic B cells (B220⁺), CD11b⁺ cells, and CD4⁺/CD8⁺ T cells. As noted in the text, there is an increase in B cell and CD11b⁺ cell frequencies and a reduction in T cell frequencies in TCox10^{-/-} and TCox10^{-/-}/Aox mice. These values are presented as frequencies of live splenocytes. The observed shifts reflect changes in population balance, not total splenic cellularity. As seen below, all genotypes have similar splenic and thymic cell counts. We have clarified this point in the revised text and added these values as a summary in the legend for Fig. 7.

For the immunization experiments shown in Figure 7D, TNP-specific IgG1 responses were measured at 2 weeks after primary immunization and 1 week after the secondary immunization (i.e., 5 weeks after the first dose, 7 days after the rechallenge). We have added these time points to the Methods section and Figure 7 legend for clarity. Regarding the Y-axis, “relative” values represent absorbance from antigen-coated wells minus background absorbance (from wells with no serum) where WT OD values were normalized to 1. This has now been clarified in the revised figure legend to avoid confusion.

(6) In figure 6F, the authors show Ag-specific T cells after infection. The data are not really convincing, based on the graph at the right it seems that there is not really a difference between the various groups of mice so it is not clear whether this is rescued. Do the authors have a non-infected mouse staining control? The frequency of cells within the regions should be indicated.

Response: We thank the reviewer for this comment. The antigen-specific T cell data referenced are shown in Figure 7H now. This panel shows CD8⁺ T cells specific for the influenza NP epitope (NP366–374) detected using H-2D^b tetramers following X31/PR8 challenge. The number of tetramer-positive cells in TCox10^{-/-} trends lower ($p = 0.098$), which is restored in TCox10^{-/-}/Aox mice (vs. WT, $p = n.s.$). Furthermore, the ability to clear lethal PR8 challenge as part of a memory T cell response that focuses on NP is enhanced in TCox10^{-/-}/Aox mice (Fig. 7I). Additionally, we have added the frequency values within the tetramer-positive gates in the figure panel. A mouse that was not immunized with X31 has been included in each staining batch to confirm tetramer specificity and establish background (Fig. 7I).

(7) figure 6H: were the cells gated on CD45.2 cells? The authors mention that they want to test whether the effect was cell-autonomous (T cell autonomous?). Since the authors transferred BM cells (and all express AOC from the Rosa26 locus), it doesn't formally exclude that other hematopoietic cells (that all express AOX) contribute to the rescue of the observed phenotypes. Perhaps I misunderstood the experiment - a better rationale/description from lines 284 on would be helpful to understand this better. Was there a difference in the reconstitution efficiency? And when were the mice infected after BM reconstitution? This is not mentioned.

Response: We thank the reviewer for pointing this out. The bone marrow chimera data originally shown in Figure 6H have been removed from the revised manuscript due to inconsistencies stemming from a technical issue with the irradiator used during those experiments. Given this, we chose to focus our conclusions on the in vivo infection model using X31/PR8, which remains robust and reproducible. To still address the question of T cell-intrinsic effects, we have added a new experiment (now shown in Figure 7E–F) in which naïve CD8⁺ OT-I⁺ T cells from different genotypes were adoptively transferred into CD45.1⁺ recipient mice and stimulated in vivo with ovalbumin. Proliferation was tracked via CellTrace Violet dilution and gating was performed on CD45.2⁺ CD8⁺ T cells, allowing us to isolate the donor population and assess genotype-intrinsic proliferation capacity. These data show a proliferation defect in TCox10^{-/-} CD8⁺ T cells that is partially rescued in TCox10^{-/-}/Aox cells, supporting a T cell-autonomous role for AOX in proliferation.

(8) The information provided in the method section is sometimes very limited. I think it is important to indicate how many cells were stimulated, in which medium and in which volume,

cytokine concentration, etc. (e.g. methods in line 368). The authors refer often in the method section to the text in the results, however it would be better to provide details in the methods as well.

Response: *We thank the reviewer for this comment. We have revised the Methods section to expand our level of detail. In addition, we have added more detail to the figure legends to help guide the reader.*

minor issues:

line 287: the word “added” should be removed.

Response: *The word “added” has been removed in the edits.*

line 481: it is not stated what the control coating was.

Response: *For the ELISA assay to determine T dependent antigens, the control wells were coated with the same antigen but received no serum. This has been clarified in the legend for Figure 7.*

Reviewer #2 lymphocyte metabolism (Remarks to the Author):

Comments:

The manuscript by Tarasenko et al., entitled ‘Cytochrome c oxidase dependent respiration is essential for T cell activation, proliferation and memory formation’ investigates the role of mitochondrial complex IV in T-cell biology. The authors explore a combination of in vitro and in vivo assays via use of COX-deficient T-cells in combination with an alternative oxidase (AOX) model to bypass the deficiency. The authors introduce the topic well, especially with regards to highlighting the requirement for OxPhos upon T-cell activation – something often overlooked in favour of the Warburg effect. Key results demonstrate that loss of COX leads to heightened apoptosis and metabolic dysregulation with AOX rescuing these phenotypes. The authors demonstrate in an infection model that loss of COX impairs the immunization and challenge response, largely attributed to loss of mitochondrial respiration. The manuscript is novel and timely, delineating the role of CIV in T-cell biology; however, I have a number of concerns that would need to be addressed.

Major

1. TMRE limitations. Accumulations of the probe are known to be affected by mitochondrial mass amongst other factors. If the authors normalised the TMRE to MitoGreen content would the difference still remain in the TCox10^{-/-} group? Could it be that the observed TMRE increase is just a result of an increased mitochondrial content. Controls are also lacking in these experiments (e.g., FCCP). Furthermore, there are multiple issues using TMRE, such as its accumulation is affected by the plasma membrane’s potential – have the authors considered whether there would be a difference here. Flow cytometry, as employed, will not tell one where

the probe is accumulating (cytosol vs mitochondria). These points may impact on conclusion draw within the paper.

Response: *While we acknowledge the limitations of TMRE and flow cytometry for assessing mitochondrial membrane potential ($\Delta\Psi_m$), normalization to MitoTracker Green presents its own challenges. MitoTracker Green is membrane potential-independent but can be influenced by mitochondrial protein content and is variably retained in damaged or dysfunctional mitochondria, limiting its reliability as a proxy for mitochondrial mass (PMID: 21595013). Given these limitations, we interpret our TMRE data in the context of our electron microscopy findings, which demonstrate a reduced number of mitochondria per field in TCox10^{-/-} T cells (Fig. 2A–E), suggesting that we may underestimate changes in $\Delta\Psi_m$. Controls with FCCP have been added to the figure. We agree that TMRE accumulation can, in some cases, be influenced by plasma membrane potential. However, our staining conditions used low, non-queching TMRE concentrations (100 nM) that preferentially reflect mitochondrial membrane potential, combined with our FCCP control as per the reviewer's suggestion.*

2. Mitochondrial morphology – linking to the point above – are more comprehensive morphological assessment is required. Given the impact on OxPhos, it would be of interest to determine whether the mitochondrial shape, number of cristae etc are impacted by TCox10^{-/-} and TCox10^{-/-}/AOX.

Response: *We appreciate the reviewer's suggestion. In response, we performed a detailed morphological analysis of mitochondria using electron microscopy. As described in the revised manuscript (Fig. 2A–E), we quantified mitochondrial length, area, width, and number per cell. TCox10^{-/-} cells showed abnormal elongated mitochondria with increased surface area and reduced number, which were restored in TCox10^{-/-}/Aox cells to a morphology resembling WT. These findings demonstrate that AOX expression rescues mitochondrial structural defects associated with COX deficiency.*

3. 2-NBDG has been demonstrated not to be a reliable indicator of glucose uptake (PMID: 32879737). Please remove all data using 2-NBDG. Alternatively, the authors should use a glucose measurement kit to demonstrate enhanced glucose uptake or use their mass spectrometry data to identify increased intracellular glucose levels if it is within the detection limits.

Response: *The 2-NBDG data have been removed from the paper. To more comprehensively assess glycolysis, we have performed additional extracellular flux analyses, including both the Glycolytic Stress Test, along with direct quantification of lactate in the culture media. These data show that both TCox10^{-/-} and TCox10^{-/-}/Aox T cells exhibit elevated glycolytic activity compared to WT (Figure 4A–D). Despite the restoration of mitochondrial respiration in TCox10^{-/-}/Aox cells, glycolysis remains elevated, suggesting that respiration alone is insufficient to restrain glycolytic flux. We hypothesize that this is due to AOX bypassing Complex III, a known source of mitochondrial reactive oxygen species (mtROS) that contribute to metabolic signaling. To test this, we treated TCox10^{-/-}/Aox T cells with MitoPQ, a mitochondrially targeted ROS donor. This intervention reduced glycolytic activity (Figure 4E), supporting the conclusion that redox signaling, likely from Complex III, is required to suppress glycolysis during T cell activation.*

Minor

1. Introduction: Some key references are missing regarding CIII and T-cell biology (PMIDs: 30626970, 23415911).

Response: *These references have been added.*

2. Given the observed differences in translation in TCox10^{-/-}/AOX cells, are the cells larger via FSC-A? If so, can the authors confirm whether normalisation to cell size has been performed on mitochondrial parameters? Furthermore, the authors could confirm whether translation is globally enhanced by performing a puromycin incorporation assay between the three/four groups.

Response: *We thank the reviewer for these comments. TCox10^{-/-} and TCox10^{-/-}/Aox cells are not larger according to FACS analysis and gated on the same region.*

A Protein Synthesis Assay kit (Click-iT, Thermo) was used to evaluate protein synthesis. The Click-iT protein synthesis assay uses a methionine analog HPG (homopropargylglycine) to label newly synthesized proteins with measurement by flow cytometry. These analogs are incorporated into proteins during active protein synthesis. TCox10^{-/-} cells have significantly less protein synthesis.

3. Line 169 – please alter the word uptake to production.

Response: *The text has been edited.*

4. Figure 2H – please demonstrate where the injections are added on the plot.

Response: *The injections have been added to the Seahorse plots.*

5. What do the corresponding ECAR plots look like for Figure 2H? Do they agree with the stable isotope tracing analysis in Figure 3? One would presume that the TCox10^{-/-} ECAR would be lower.

Response: *We appreciate the reviewer's question and agree that mitochondrial dysfunction is often compensated by increased glycolytic flux. In line with this, TCox10^{-/-} T cells exhibit elevated ECAR, as shown in Fig. 4A, indicating a glycolytic shift despite impaired oxidative phosphorylation. This elevated ECAR is maintained in TCox10^{-/-}/Aox cells, demonstrating that AOX expression does not suppress glycolysis. Media lactate levels are similarly elevated (Fig. 4C).*

Although ¹³C-glucose labeling of intracellular lactate is reduced in TCox10^{-/-}/Aox cells (Fig. 4E), this does not reflect impaired glycolytic flux. Instead, it is consistent with enhanced lactate export, leading to lower intracellular labeling despite preserved extracellular accumulation. We have corrected this in the manuscript.

To clarify the mechanism sustaining this glycolytic phenotype, we used the mitoPQ system to selectively restore mitochondrial ROS signaling via superoxide. As shown in Fig. 4D, mitoPQ suppressed ECAR, supporting the model that loss of mitochondrial ROS signaling contributes to elevated glycolysis in AOX-expressing cells.

6. Were the stable isotope tracing experiments performed in the presence of 10% dialysed FBS? This should be stated clearly in the methods rather than reference to previous publications.

Response: *We agree that the use of dialysed FBS can be helpful in some isotope tracing contexts. However, in this study, we used regular FBS. While this may introduce some background metabolites, our analysis focused on relative labeling differences across similar conditions. We have now clearly stated in the Methods that standard (non-dialysed) FBS was used.*

7. Do the authors see increased reductive carboxylation in the TCox10^{-/-} T-cells as a potential compensatory mechanism to CIV loss? If so, this should be included and discussed.

Response: *In our previous paper (PMID: 28591633), we reported that we did not observe reductive carboxylation in TCox10^{-/-} T cells by stable isotope studies with ¹³C-glutamine.*

8. Figure 5A – only a representative plot is shown, can the authors please add the formal proliferation analysis.

Response: *A formal proliferation analysis has been added and is now Fig. 6A.*

9. Representative plots are missing for Figure 5B.

Response: *These panels have been removed and representative plots have been added to Fig. S4.*

10. Concentrations of probes lacking in methods, particularly in the flow cytometry section.

Response: *We have revised the methods as well as the figure legends to improve clarity.*

Reviewer #3 mitochondrial metabolism (Remarks to the Author):

The current study determines the effects of restoring respiratory flux in Cox10 deficient mice by expressing the alternative oxidase (AOX), which enables restoring electron transfer flux initiated at complex I and complex II. Authors find that most of the metabolic and functional changes induced by Cox10 deficiency are reversed by AOX expression. The major concerns stem from a misunderstood gap in knowledge that this paper claims to cover, as well as some of the conclusions that are not supported by the data. More specifically:

1) Authors use a statement in the introduction that leads to the interpretation that the major and direct action of complex IV (COX) is to regulate apoptosis, mtDNA maintenance and transcription. It has been widely described for mitochondrial diseases and in many different tissues, including by using AOX, that a defect in electron transport chain activity is the primary event leading to secondary changes in mtDNA maintenance, transcription (i.e. mtISR) and cell death. The text leads to the wrong impression that this is different in lymphocytes and that this study addresses such an inexistent gap. This study only confirms the same connections that have been observed in other tissues, but now in lymphocytes. Authors should tone down claims of novelty regarding that this is the first study testing the connection between respiration, apoptosis and transcription.

Response: *We thank the reviewer for their comment and have revised the manuscript to sharpen the mechanistic focus of our study. While mitochondrial respiration has been broadly studied in various cell types, metabolic requirements and regulatory mechanisms are highly context-dependent and not universally conserved across all lineages. The central finding is that mitochondrial respiration is the necessary and sufficient function of COX required for T cell activation, proliferation, and differentiation. Although much of the immunometabolism field has emphasized glycolysis during early T cell activation, our data establish that mitochondrial respiration plays an essential and instructive role. To directly dissect the contribution of respiration independent of proton translocation by COX, we used TCox10^{-/-} T cells expressing alternative oxidase (AOX), which restores electron transport to oxygen without pumping protons. This genetic model allowed us to isolate mitochondrial respiration from COX-*

dependent ATP synthesis. AOX expression rescued oxygen consumption, mitochondrial ATP levels, redox balance, and membrane potential, and was sufficient to restore viability, proliferation, differentiation into effector and memory subsets, and antiviral immunity (Figures 2, 5, and 6).

We further demonstrate that sustained electron input into the ETC is required to support AOX-mediated mitochondrial membrane potential and ATP production, with Complex I playing the primary role and Complex II and dihydroorotate dehydrogenase (DHODH) providing supportive contributions (Fig. 3D). Inhibition of any of these upstream inputs impaired proliferation (Fig. 6C), indicating that ETC-linked dehydrogenases are not functionally interchangeable in this context. Thus, AOX permits us to define respiration, rather than proton translocation, as the essential output of COX in T cells.

Despite restored respiration, glycolytic flux remained elevated in *TCox10^{-/-}/Aox* cells. Treatment with MitoPQ, a mitochondrially targeted ROS donor, suppressed glycolysis (Fig. 4D), indicating that redox signaling, likely from Complex III, is required to restrain glycolysis and support metabolic reprogramming. These results show that respiration and redox signaling are separable functions within the ETC, both required for proper immune activation.

These findings highlight the importance of experimentally defining mitochondrial requirements in specific cell types and contexts rather than relying on published studies in other cell types. As shown below, COX-deficient B cells show comparable levels of apoptosis to WT (panel A), in contrast to the apoptotic response observed in COX-deficient T cells. However, this does translate into lower rates of proliferation (WT – blue, *BCox10^{-/-}* – Red) as determined by cell trace violet (panel B). These data reinforce our assertion that cell specific contexts are critical.

These observations also distinguish immune cell metabolism from that of transformed cells. Immunometabolism originated from studies of cancer metabolism, where many principles, such as flexible electron input and high reliance on glycolysis, were first defined. However, our

findings demonstrate that these principles do not fully apply to primary immune cells. In contrast to tumor models (PMID: 32641834, 35346867, 30449682, 17967865) T cells require a defined mitochondrial circuit with specific electron donors and redox control mechanisms.

Thus, our study shows that respiration is the key functional output of COX in T cells, and that defined ETC inputs and redox signaling, not just ATP generation, are required to support immune fate and function. This work reinforces the need for mechanistic, context-specific evaluation of mitochondrial pathways in immunology.

2) Is the activation of the mitochondrial ISR involved in the defect in proliferation and cell death? How many of the cell death genes identified in the RNAseq converge with the mitochondrial ISR? Mitochondrial respiratory dysfunction, depolarization, decreased ATP/ADP and elevated ROS activate OMA1 protease, which activates HRI to block protein translation and activate ATF4 to induce cell death.

Response: *We appreciate the reviewer's interest in the integrated stress response (ISR). While mitochondrial stress can activate the ISR, our data suggest that the primary mechanism of cell death in COX-deficient T cells is mediated through the Fas/FasL pathway. As shown in Figure 5D–E, AOX expression reduces Fas and FasL expression and prevents activation-induced cell death as a primary mechanism (Extended Data Figure 3B–C). We also observed increased ATF4 expression in our RNAseq dataset (Extended Data Fig. 3A), consistent with ISR activation. The possibility that ISR signaling could influence Fas/FasL expression is intriguing and subject to future study.*

3) Authors conclude that reductive stress and ROS production are the major drivers for the defects induced by Cox10 deletion. However, no direct evidence is provided to support this claim, as T-cell death and lack of proliferation could also be a result of decreased mitochondrial ATP synthesis or activation of the mitochondrial ISR. To draw such a conclusion, authors should induce reductive stress in Cox10^{-/-} AOX expressing lymphocytes and determine whether it reverses the benefits induced by AOX expression, via the expression of EcSTH for instance. ATP/ADP should be provided as well to determine whether increased glycolysis to lactate is not effective in restoring this ratio and rather decreased ATP is only reflective a decrease in the nucleotide pool destined for DNA replication.

Response: *We appreciate the reviewer's thoughtful critique and the opportunity to clarify our mechanistic model. While we agree that multiple pathways may contribute to T cell dysfunction following COX deletion, our data demonstrate that mitochondrial respiration, rather than proton pumping, is necessary and sufficient for T cell activation, proliferation, and differentiation.*

In response to the suggestion to use EcSTH to model reductive stress, this approach is not feasible in our system. The key period of metabolic reprogramming in T cells occurs within the first 24 hours of activation. EcSTH requires transfection and sustained expression, which would not align temporally with this early activation window. Thus, it cannot be deployed within the critical timeframe necessary for probing these events.

Instead, we directly tested the requirement for respiration using a genetic approach. By expressing AOX in TCox10^{-/-} T cells, we selectively bypassed the proton-pumping function of COX while maintaining electron flow to oxygen. This allowed us to isolate the role of mitochondrial respiration from ATP synthase-driven proton gradient utilization. AOX restored oxygen consumption, ATP levels, mitochondrial membrane potential, and redox balance (Figures 2 and 3). Crucially, it also rescued proliferation, survival, and effector/memory T cell differentiation (Figures 5 and 6), demonstrating that mitochondrial respiration, independent of COX-mediated proton translocation, is necessary and sufficient to restore T cell function.

To determine which upstream electron sources are required to sustain ETC function when respiration is maintained by AOX, we inhibited Complex I (piericidin), Complex II (TTFA), and DHODH (brequinar). Piericidin and TTFA treatments led to drops in TMRE staining and ATP (Figure 3D–E), indicating that Complex I is the primary source of electron input for maintaining mitochondrial membrane potential and ATP production, with Complex II playing a supportive role. DHODH inhibition modestly reduced TMRE under certain conditions but had minimal impact on mitochondrial ATP, suggesting its contribution lies more in redox balance and biosynthesis than core energetics. Despite these differences, inhibition of any of these dehydrogenases impaired proliferation in AOX-expressing TCox10^{-/-} T cells (Figure 6C), demonstrating that continuous electron input from multiple upstream sources is necessary to maintain the ETC circuit and support T cell proliferation, even when respiration proceeds independently of proton pumping.

In sum, our data support a model in which mitochondrial respiration, driven by specific electron donors into the ETC, is both necessary and sufficient for T cell function. The benefits of AOX are not due to proton translocation, but rather due to restoration of a functional respiratory circuit that enables redox signaling, metabolic fitness, and proliferative capacity.

4) Electron microscopy or confocal imaging is needed to establish how much of the changes in the multiple fluorescence probes used by flow cytometry are reporting on membrane potential and ROS rather than just mitochondrial mass. The increases and trend to increase in mitoPY and TMRE fluorescence reported might be just reflecting the increase in mitochondrial mass induced by Cox10, as observed in muscles from patients with MERFF. Thus, when normalized by mitochondrial mass, it is likely that membrane potential and H₂O₂ are decreased in Cox10^{-/-} cells. High-resolution live cell imaging is also needed to confirm that the concentrations of mitosox used for flow cytometry only stain mitochondria and not the nucleus.

Response: *We appreciate the reviewer's detailed comment and agree that changes in mitochondrial mass and morphology can influence the interpretation of flow cytometry data using mitochondrial probes such as TMRE, mitoPY, and MitoSOX. To address this, we did not normalize these probes to MitoTracker Green, given its known limitations in dysfunctional mitochondria (PMID: 21595013). Instead, we relied on electron microscopy (Fig. 3A–D), which demonstrates that TCox10^{-/-} T cells have fewer mitochondria/field and reduced total mitochondrial area compared to WT, despite being more elongated. This finding argues against increased mitochondrial mass as the source of elevated mitoPY and TMRE signals.*

Given this reduced mitochondrial content, the observed increases in mitoPY, TMRE, and MitoSOX fluorescence likely underestimate the degree of mitochondrial dysfunction on a per-organelle basis. These interpretations are further supported by independent functional readouts, including increased total ROS, mitochondrial hyperpolarization, and disrupted NAD⁺/NADH balance.

Regarding the specificity of MitoSOX, we used concentrations optimized to selectively stain mitochondria and not the nucleus, as recommended in manufacturer protocols. We acknowledge that flow cytometry cannot directly confirm subcellular localization, and we appreciate the reviewer's suggestion. However, the MitoSOX concentrations and staining conditions used in our experiments are based on validated protocols that minimize nuclear signal and are optimized for selective mitochondrial detection.

5) Based on transcriptional analysis and % labeled lactate from glucose, authors claim that glycolysis is decreased in Cox10^{-/-}. However, it is widely known that defects in mitochondrial ATP synthesis are compensated by an increase in glycolysis to lactate, otherwise there is a severe ATP deficiency and NADH accumulation. Thus the decrease in transcript levels might just be reflecting a decrease in proliferative capacity, rather than lower glycolytic flux per se. How do the ECAR in response to oligomycin and lactate excretion rates look like in Cox10^{-/-} cells versus Cox10^{-/-}/Aox cells, basally and when these cells are stimulated to proliferate? Can authors measure extracellular lactate content to determine whether there is increased excretion?

Response: *We agree with the reviewer that mitochondrial dysfunction is often compensated by increased glycolytic flux, and have performed additional studies to clarify this point. Indeed, TCox10^{-/-} T cells exhibit a markedly elevated ECAR, as shown in Fig. 4A, and this elevated glycolysis is maintained in TCox10^{-/-}/Aox cells, indicating that AOX expression does not suppress the glycolytic phenotype. Furthermore, lactate in the media also remains elevated (Fig. 4C). Although ¹³C-glucose labeling of intracellular lactate is reduced (Fig. 4E), this is not due to impaired glycolysis. Instead, our new findings are consistent with rapid lactate export. To clarify why glycolysis remains elevated in TCox10^{-/-}/Aox cells, we used the mitoPQ system to restore mitochondrial ROS signaling with superoxide. As shown in Fig. 4D, MitoPQ suppresses glycolysis, providing a mechanism by which glycolytic phenotype persists due to the absence of mtROS.*

REVIEWERS' COMMENTS

Reviewer #1 (Remarks to the Author):

The authors responded well to my comments.

Response: We appreciate the comments and suggestions of Reviewer #1.

Reviewer #2 (Remarks to the Author):

The authors have addressed my comments in full. Congratulations on an interesting study.

Response: We appreciate the comments and suggestions of Reviewer #2.

Reviewer #3 (Remarks to the Author):

The authors addressed all the comments and the main conclusions are now supported by data. I only have minor suggestions for the discussion:

-Authors show that ATF4 targets are engaged and thus the mitoISR. It could be discussed that this ISR activation could be contributing to the decrease in proliferation, as ISR blocks protein translation.

Response: We appreciate this important insight. Indeed, we did see reduced protein synthesis, as requested by another reviewer. We appreciate the suggestion and will include this in the discussion.

- The mitoPQ result is puzzling. MitoPQ increases the production of superoxide in complex I. This superoxide needs SOD2 activity to generate H₂O₂ and then induce signaling, as proposed by the authors. The expectation is that mitoPQ would further damage Tcox10 ^{-/-} mitochondria, as the authors show that complex I activity is needed for AOX to rescue and mitoPQ is expected to damage complex I. A discussion reconciling the established mechanism of mitoPQ action and the effects observed needs to be rewritten, as it seems unlikely that mitoPQ is changing glycolysis by increasing H₂O₂ release from mitochondria.

Response: We thank the reviewer for raising this important point. We have removed the MitoPQ data from the manuscript, as this is part of an ongoing investigation in our laboratory into the regulation of glycolysis by OXPHOS. We agree that the mechanism by which glycolysis remains elevated in AOX-rescued cells is not resolved. While one possibility is altered redox signaling due to bypass of Complex III, other explanations cannot be excluded. We have therefore modified the Discussion to present this as an open question and to highlight it as a direction for future investigation, rather than attributing the effect to a specific mechanism.

-I would correct the statements of energy production across the manuscript. Energy is not produced by mitochondria. ATP or heat are produced by mitochondria by transforming the energy contained in nutrients.

Response: We appreciate the precision of this comment. Reviewer #3 is quite correct. We have revised the manuscript to correct statements that previously referred to “energy production” by mitochondria. As the reviewer notes, mitochondria do not produce energy de novo. Rather, they transform the chemical energy contained in nutrients into ATP or heat.